



# An improved hydro-biogeochemical model (CNMM-DNDC V6.0) for simulating dynamical forest-atmosphere exchanges of carbon and evapotranspiration at typical sites subject to subtropical and temperate monsoon climates in eastern Asia

Wei Zhang[1, 2], Xunhua Zheng[1,2,3], Siqi Li[1], Shenghui Han[1], Chunyan Liu[1,3], Zhisheng Yao[1,3], Rui Wang[1], Kai Wang[1], Xiao Chen[1], Guirui Yu[4,5], Zhi Chen[4,5,6], Jiabing Wu[7], Huimin Wang[4], Junhua Yan[8], Yong Li[1]

[1] State Key Laboratory of Atmospheric Boundary Layer Physics and Atmospheric Chemistry, Institute of Atmospheric Physics, Chinese Academy of Sciences, Beijing 100029, P. R. China

[2] Qilu Zhongke Institute of Carbon Neutrality, Jinan 250100, China

[3] College of Earth and Planetary Sciences, University of Chinese Academy of Sciences, Beijing 100049, P. R. China

[4] Key Laboratory of Ecosystem Network Observation and Modeling, Institute of Geographic Sciences and Natural Resources Research, Chinese Academy of Sciences, Beijing 100101, China

[5] College of Resources and Environment, University of Chinese Academy of Sciences, Beijing 100049, China

[6] Beijing Yanshan Earth Critical Zone National Research Station, University of Chinese Academy of Sciences, Beijing 101408, China

[7] Institute of Applied Ecology, Chinese Academy of Sciences, Shenyang 110016, China

[8] South China Botanical Garden, Chinese Academy of Sciences, Guangzhou 510650, China

*Correspondence to*: Siqi Li (lisiqi@mail.iap.ac.cn); Yong Li (yli@mail.iap.ac.cn)

**Abstract.** Carbon exchange between forest ecosystems and the atmosphere play an important role in global carbon cycle, which is difficult to be accurately quantified due to the large uncertainties in scaling up site-scale observations or filling-up measurement gaps. A process-oriented model equipped with comprehensive processes to explicitly simulating coupled carbon, nitrogen and water cycling, is hypothesized to reduce the uncertainties in quantification of forest carbon fluxes. To test this hypothesis, the Catchment Nutrient Management Model - DeNitrification-DeComposition (CNMM-DNDC), as a hydro-biogeochemical model that dynamically couples the carbon, nitrogen, phosphorous and water cycling processes, was updated in this study by incorporating a new forest growth module derived from the Biome-BGC model and validating the updates using multiple-year continuous observations of carbon and water fluxes at the site scale. The updated model has





improved the processes of photosynthesis, litter decomposition, allocation, respiration and mortality to more effectively
capture the transformation and transportation of nutrients in plant-soil-water continuum. The observed gross primary
productivity (GPP), ecosystem respiration (ER), net ecosystem carbon dioxide exchange (NEE) and evapotranspiration (ET)
of three typical forest sites subject to subtropical and temperate climates in eastern Asia (2003–2010) were used for the
model calibration and validation. Compared with the original model in validation, the updated model showed significant
improvements in simulating the daily dynamics and inter-annual variations of each variable, with the NRMSE values
decreased by 46% and 54%, 65% and 37%, 4% and −6%, and 38% and −3% for GPP, ER, NEE, and ET on daily and annual
scales, respectively. The comparable performances of both model versions for annual NEE emphasizes the importance of
validating each component of carbon fluxes to avoid the offsetting of model errors. The canopy average specific leaf area,
fraction of leaf nitrogen in Rubisco, annual leaf and fine root turnover fraction, maximum stomatal conductance and the ratio
of carbon to nitrogen in leaves and fine roots were identified as the sensitive eco-physiological parameters affecting the
simulations of GPP and ER. In addition, the meteorological variables of solar radiation, humidity and air temperature also
showed strong influences on the simulated GPP and ER. The relatively satisfactory performances demonstrated that the
modified model has the ability to capture the daily dynamics and inter-annual variations of carbon fluxes for forests in
temperate and subtropical zones, which is essential for estimating the emissions of greenhouse gases at the regional or global
scales.
**1 Introduction**
Forest ecosystems are widely spread in the world, which accounts for nearly 40% of the Earth's ice-free land surface
(Waring and Running, 1998). In spite of providing wood products, forests can also prevent soil erosion, maintain
biodiversity and regulate the water and carbon cycles (Waring and Running, 1998; Chiesi $et\ al.$, 2007). Forests play an
important role in global carbon cycle by regulating atmospheric carbon dioxide ($CO_2$) level via the photosynthesis. In
combating with climate change, forest can remove approximately one quarter of $CO_2$ emitted by combustion of fossil fuels
and industry through carbon sequestration (Seidl $et\ al.$, 2017; Cook-Patton $et\ al.$, 2020; FAO, 2020). Hence, accurate
estimation of forest carbon fluxes, such as gross primary productivity (GPP) and ecosystem respiration (ER), is essential for



understanding terrestrial ecosystem carbon cycle and stabilizing global climate change (Mao *et al.*, 2016; Ren *et al.*, 2022).
Various methods can be applied to estimate the carbon fluxes of forests, such as eddy-covariance techniques, satellite-based
remote sensing and numerical models. In comparison, the numerical models are promising tools to combine data from
different sources for more complete characterization of vegetation and soil processes. Among different types of numerical
models, including statistical/regression models (Tatarinov and Cienciala, 2006; Raj *et al.*, 2014), light use efficiency models
(Running *et al.*, 2004; Yuan *et al.*, 2014) and process-oriented models, the last type is essential scientific tools that, deal with
the atmosphere, vegetation, soil, and water within a given space as a continuous and dynamic system (Makela *et al.*, 2000;
Mao *et al.*, 2016). The process-oriented models are generally established based on theoretical understandings of
biogeochemical or even hydro-biogeochemical processes, and thus can better estimate the terrestrial carbon budget
influenced by multiple factors (Schulze *et al.*, 2009; Friedlingstein and Prentice, 2010; Hidy *et al.*, 2012).
Some process-oriented models have been developed to simulate carbon cycle of natural/undisturbed forest ecosystems
at regional or global scale, such as Biome-BGC, ORCHIDEE and LPJ (White *et al.*, 2000; Sitch *et al.*, 2003; Krinner *et al.*,
2005). As managed ecosystems are playing essential roles in terrestrial carbon budget, more researches have most recently
focused on the managed ecosystems with significant anthropogenic activities, such as afforested forests and croplands (Cai
*et al.*, 2022; Ciais *et al.*, 2013; Liu *et al.*, 2022; Mao *et al.*, 2016; Miyauchi *et al.*, 2019). Meanwhile, it is now clear that
interactions between climate and carbon cycle are affected by nitrogen availability and hydrology processes in ecosystems
(Thornton *et al.*, 2007; Piao *et al.*, 2013). The consideration of carbon-nitrogen interactions, as well as nutrients limitations
and availability and/or dynamics of water, in the process-oriented model substantially changes simulated dynamics of several
critical feedbacks between land and climate systems (e.g., Churkina *et al.*, 2009; Thomas *et al.*, 2013). Thus, the explicit
simulations of the nitrogen cycle and the interactions among carbon, nitrogen and water cycles have been improved in some
biogeochemical models. For instance, the widely applied Biome-BGC has been improved to create Biome-BGCMuSo by
adding a multilayer soil module, implementing the processes related to soil moisture and nitrogen balance, and incorporating
the vegetation managements suitable for managed forests and croplands (Thornton *et al.*, 2002; Bond-Lamberty *et al.*, 2005;
Tatarinov and Cienciala, 2006; Di Vittorio *et al.*, 2010; Hidy *et al.*, 2016). Other models suitable for undisturbed ecosystems
have also been extended to simulate the interactions of carbon, nitrogen and water in the managed ecosystems (e.g.,



croplands) with higher accuracy, such as ORCHIDEE-STICS and LPJmL (Gervois *et al.*, 2004; Bondeau *et al.*, 2007). For
croplands, some other process-oriented models have also been established to simulate the complex biogeochemical processes,
such as DNDC, WNMM, SPACSYS and WHCNS (Li *et al.*, 1992; Li, 2007; Wu *et al.*, 2007; Liang *et al.*, 2016). The
DNDC model has been extended to PnET-N-DNDC and Wetland-DNDC versions, which are suitable for forest and wetland
ecosystems (Li *et al.*, 2000; Zhang *et al.*, 2002). The WNMM model has been updated to the CNMM model by incorporating
the new hydrological module and improving plant growth module fitted for different types of vegetation (Ma *et al.*, 2018).
As the biogeochemical processes closely link with the hydrological processes, especially for the landscapes with undulating
terrain (e.g., catchments or river basins), hydro-biogeochemical models are believed to be more effective tools for studying
the carbon, nitrogen and water cycles for terrestrial ecosystems. Generally, hydro-biogeochemical models are based on
distributed or semi-distributed hydrological models and biogeochemical models with different levels of complexity, such as
LandscapeDNDC-CMF, SWAT-DayCent and CNMM-DNDC (Haas *et al.*, 2012; Wu *et al.*, 2016; Zhang *et al.*, 2018).
The CNMM-DNDC is a hydro-biogeochemical model. It was initially developed by fully coupling the Catchment
Nutrient Management Model (CNMM) and the DeNitrification-DeComposition Model (DNDC) and then further updated
several times to enable comprehensive simulations of the tightly coupled carbon, nitrogen, phosphorous and water cycles at
site, catchment, river basin, regional and even continental scales (Zhang et al., 2018; Zhang et al., 2018; Zhang et al., 2021a;
Zhang et al., 2021b; Li et al., 2022; Li et al., 2023). This model has been applied in different catchments located in various
climate regions by simultaneously simulating the ecosystem productivity, emissions of greenhouse gases and gaseous air
pollutants, and hydrological nitrogen losses through soil leaching and discharge in streams as well as soil erosion from an
entire catchment or individual landscape units (Zhang *et al.*, 2018; Zhang *et al.*, 2021a; Zhang *et al.*, 2021b; Li *et al.*, 2022; Li
*et al.*, 2023). At present, this model has been validated with multiple simultaneously variables, including crop yields, soil
organic carbon contents, emissions of methane, nitrous oxide, nitric oxide and ammonia, hydrological losses of nitrate and
total nitrogen through surface runoff and stream flows, and sediments production in soil erosion, in a subtropical agro-forest
catchment (Zhang *et al.*, 2018; Li *et al.*, 2023). The comprehensive validation demonstrated the strong capacity of the
CNMM-DNDC model in simulating the hydro-biogeochemical processes of terrestrial ecosystems.





Despite the sufficient validations in croplands, the model performance in simulating the forest ecosystems has not been
comprehensively evaluated due to limited data. It is very necessary to assess the model performance in the forest ecosystems
due to the vital role of forests in the terrestrial carbon budget (Seidl *et al.*, 2017). In addition, the present CNMM-DNDC
model simulates vegetation growth by adapting the module applied in the BIOME3 model, which sets the plants as an
entirety without biomass allocation among different tissues and/or organs. Such simplification may induce large
uncertainties, as both photosynthetic allocation and mortality are substantially important for accurately simulating the carbon
and nitrogen cycles of forest ecosystems. Therefore, we hypothesize that modification of the current vegetation growth
module by improving the simulation mechanisms of related processes, such as biomass allocation, respiration, and mortality,
can improve the performance of the CNMM-DNDC in simulating the carbon cycle of forest ecosystems in different climate
regions. To achieve such improvements is especially necessary to broaden the applicability and to increase the reliability of
this model.
To test the above hypothesis, the original and modified versions of the CNMM-DNDC model were evaluated for their
simulation of the daily GPP, ER, net ecosystem carbon dioxide exchange (NEE) and evapotranspiration (ET) using the
continuously observed multiple-year data of three typical forests subject to subtropical and temperate monsoon climates in
the eastern Asia. The overarching goals of this study are to (i) attempt to fill the gaps of simulation mechanisms in the
CNMM-DNDC model by improving the scientific processes of vegetation growth, (ii) compare the performances of the
original and modified model versions in simulating the GPP, ER, NEE and ET at the three typical forest sites and, (iii)
identify the eco-physiological parameters and model inputs that are sensitive to the simulated GPP and ER of the examined
forests. The new model version with the validated modifications is expected to be applicable for more accurately estimating
the carbon budget of forest ecosystems.



## 2 Materials and methods

### 2.1 Model description

#### 2.1.1 Overview of CNMM-DNDC

The initial version of the CNMM-DNDC model was established by incorporating the soil carbon and nitrogen transformation/transferring processes of the DNDC model, including the processes of decomposition, nitrification, denitrification and fermentation, into the distributed hydrological framework of the CNMM model (Zhang *et al.*, 2018). All other functions of the initial CNMM-DNDC version, such as those on the phosphorous cycle and vegetation growth, were directly inherited from the CNMM (Li *et al.*, 2017; Zhang *et al.*, 2018). Its later versions were established through several updates to improve its comprehensive functions and universally applicability. These updates include a) developing new soil pH regulation mechanisms for tea plantations (Zhang *et al.*, 2020), b) modifying the energy balance and heat conductivity mechanism for adaptation to freeze-thaw cycles in permafrost regions (Zhang *et al.*, 2021b), c) developing an alga-regulation mechanism of flooding water pH and introducing the Jayaweera-Mikkelsen mechanism for ammonia transfer between water and the atmosphere to improve the simulation of ammonia volatilization from paddy rice fields (Li *et al.*, 2022) and, d) introducing the ROSE mechanism on soil erosion to enable the simulations of hydraulic soil erosion and hydrological losses of dissolved and particle carbon, nitrogen and phosphorous components (Li *et al.*, 2023). As this model has been developed based on the basic theories of physics, chemistry, and biogeochemistry, it has realized systematic simulation of the tightly coupled carbon, nitrogen and water cycles at the catchment scale.

The simulated soil depth and temporal and spatial resolutions of the model are all allowed to user-defined depending on the availability of driving data and/or research objectives. The simulated soil profile could be down to 4 m deep. The soil temperature is calculated based on energy balance and heat conductivity. The soil moisture is simulated based on the mass balance among precipitation, irrigation, evapotranspiration, vertical flow, lateral flow and water from a rising groundwater table. For the site scale simulation, only surface runoff is taken into account, but the subsurface lateral flow is not. The infiltrated water is controlled by a defined maximum infiltration rate. The vertical water movement in the soil profile is calculated following the Darcy's law. A cell-by-cell approach using a kinematic approximation is applied to route the saturated





overland and subsurface flow based on the digital elevation model. A cascade of linear channel reservoirs is used for
calculating the stream flow (Wigmosta *et al.*, 1994). The soil biogeochemical processes are generally based on the first-order
kinetics for organic matter decomposition and the Michaelis-Menten kinetics of two substrates for nitrification and
denitrification while using the concept of an "anaerobic balloon" to allow for simultaneously oxidative and reductive
reactions (Li, 2007). For more details on the simulations of the soil biogeochemical processes, please see Li (2000, 2007), Li
*et al.* (2022, 2023) and Zhang *et al.* (2018, 2020, 2021).
In the current CNMM-DNDC model, the growth processes of plants, including crops, forests and grasslands, are adapted
from those applied in the BIOME3 model (Haxeltine and Prentice, 1996a, b). Among these processes, photosynthesis rate is
calculated as functions of absorbed photosynthetically active radiation (APAR), temperature and atmospheric $CO_2$
concentration, following Farquhar *et al.* (1980) for $C_3$ species and Collatz *et al.* (1992) for $C_4$. More explicitly, the net
photosynthesis rate, i.e., net primary productivity, is calculated using a standard nonrectangular hyperbola formulation, which
gives a gradual transition between two limiting rates describing the responses of photosynthesis to APAR and the Rubisco
abundance (Haxeltine and Prentice, 1996b). For more details on vegetation growth in the current CNMM-DNDC version,
please see Haxeltine and Prentice (1996a, b) and Zhang *et al.* (2018). The currently adopted simulation mechanisms on plant
growth, however, ignore not only the biomass allocation among different plant organs or tissues but also the processes of
mortality, even though both photosynthetic allocation and mortality are substantially important for accurately simulating the
carbon and nitrogen cycles of forest ecosystems.

**2.1.2 Improvements of CNMM-DNDC**

In this study, a new module of forest growth were established for the CNMM-DNDC model, by referring to the
simulation mechanisms applied in the Biome-Bio Geochemical Cycles (Biome-BGC) model (White *et al.*, 2000) to improve
the model performance in simulating the plant growth of forest ecosystems, as well as the related water and carbon fluxes. This
module is designed to well simulate the processes of photosynthesis, litter decomposition, photosynthetic allocation,
respiration and mortality of forests.





**2.1.2.1 Carbon pools of forest ecosystems**

Its carbon pools consist of plant carbon, coarse woody debris carbon, litter carbon and soil carbon. The plant carbon pool is divided into sub-pools of leaves, fine roots, live stems, dead stems, live coarse roots and, dead coarse roots. The coarse woody debris pool is the accepter of dead stems and dead coarse roots. The litter carbon pool is made of four constituents, which are liable carbon, unshielded cellulose, shielded cellulose and, lignin. The soil carbon pool excluding the litter carbon is initially inherited from the DNDC model, which includes the sub-pools of microbes, humads and humus.

**2.1.2.2 Photosynthesis**

The photosynthesis is one of the important individual processes for carbon accumulation in a forest ecosystem. The photosynthesis rate is simulated separately for sun and shade leaves based on the two-leaf mechanism in consideration of enzyme kinetics (Farquhar *et al.*, 1980). It jointly depends on the amount of APAR, the calculated maintenance respiration (MR), the difference between the internal and external partial pressure of $CO_2$, the nitrogen content in leaves, the portion of nitrogen in Rubisco and, the temperature controlling the enzyme kinetics.

The projected leaf area index of whole canopy ($PLAI_{total}$, in $m^2 \ m^{-2}$) is calculated as the product of average specific leaf area based on carbon mass and leaf carbon content. The projected leaf area indexes for the sun leaves ($PLAI_{sun}$, $m^2 \ m^{-2}$) and the shade leaves ($PLAI_{shade}$, $m^2 \ m^{-2}$) are calculated following Eqs. 1−2 (Jones, 1992).

$$PLAI_{sun} = 1 - e^{-PLAI_{total}} \qquad \text{Eq. 1}$$

$$PLAI_{shade} = PLAI_{total} - PLAI_{sun} \qquad \text{Eq. 2}$$

The amount of APAR ($APAR_{total}$, $APAR_{sun}$, $APAR_{shade}$, $W \ m^{-2}$) is calculated based on the incoming shortwave radiation (Rads, in $W \ m^{-2}$), albedo (Alb, dimensionless), canopy light extinction coefficient ($k$, dimensionless), the $PLAI_{total}$ and, the fraction of PAR in the incoming shortwave radiation ($f_{par}$, dimensionless) for the sun and shade leaves, respectively (Eqs. 3−5).

$$APAR_{total} = f_{par}Rads(1.0 - Alb)(1.0 - e^{-kPLAI_{total}}) \qquad \text{Eq. 3}$$

$$APAR_{sun} = f_{par}kPLAI_{sun}Rads(1.0 - Alb) \qquad \text{Eq. 4}$$

$$APAR_{shade} = APAR_{total} - APAR_{sun} \qquad \text{Eq. 5}$$





Three separate equations (Eqs. 6–8) are followed to calculate the reaction rate of photosynthesis, which is the process of
generating simple sugars by combining the $CO_2$ and $H_2O$ molecules using energy from the sun (Farquhar *et al.*, 1980).
The Eq. 6 shows the regulation of $CO_2$ diffusion on the photosynthetic rate regulated by Rubisco ($A_v$, µmol $CO_2$ m$^{-2}$ s$^{-1}$)
or by electron transfer($A_j$, µmol $CO_2$ m$^{-2}$ s$^{-1}$), wherein $C_a$ and $C_i$ (Pa) are the atmospheric and intercellular concentrations of
$CO_2$, respectively. The $C_a$ is calculated using Eq. 7, based on the input data of atmospheric pressure (pa, Pa) and $CO_2$
concentration ($C_{co_2}$, µmol mol$^{-1}$). Following Eq. 8 (Nobel, 1991), the stomatal conductance of $CO_2$ ($g_{CO_2}$, µmol m$^{-2}$ s$^{-1}$ Pa$^{-1}$)
is calculated using the conductance of water vapour ($g_{H_2O}$, m s$^{-1}$), air temperature ($T_{air}$, °C), the universal gas constant ($R$ =
8.3143, m$^3$ Pa mol$^{-1}$ K$^{-1}$) and, the ratio of molecular weights of water vapour to $CO_2$ ($M$). The $A_v$ or $A_j$ in Eq. 6 is on the PLAI
basis.

$$A_{\text{v or j}} = g_{CO_2}(C_a - C_i) \qquad \text{Eq. 6}$$

$$C_a = 10^{-6}C_{co_2}\text{pa} \qquad \text{Eq. 7}$$

$$g_{CO_2} = \frac{10^6 g_{H_2O}}{MR(T_{air} + 273.15)} \qquad \text{Eq. 8}$$

The $A_v$ is simulated by Eqs. 9–16. In Eq. 10, the maximum rate of carboxylation ($V_{max}$, µmol $CO_2$ m$^{-2}$ s$^{-1}$) is simulated as
a function of the nitrogen mass in per unit area of sun or shade leaves ($N_{leaf}$, kg N m$^{-2}$), the fraction of leaf nitrogen in Rubisco
as an input parameter ($f_{NR}$, kg N kg$^{-1}$ leaf N), the weight proportion of Rubisco relative to its nitrogen content (WP = 7.16, kg
Rub kg$^{-1}$ Rubisco N) (Fasman, 1976) and the activity of Rubisco (AC, µmol $CO_2$ kg$^{-1}$ Rub s$^{-1}$). Using Eq. 11, the nitrogen
concentration ($N_{leaf}$) in sun and shade leaves are calculated based on the input parameters of the mass ratio of carbon to
nitrogen (C:N) in leaves ($r_{cnl}$, dimensionless) and specific leaf area (SLA, m$^2$ kg$^{-1}$ C). In Eq. 12, the AC is adapted by air
temperature ($T_{air}$, °C) from its standard value at 25 °C (AC$_{25}$, µmol $CO_2$ kg$^{-1}$ Rub s$^{-1}$), wherein the AC$_{25}$ value and the
temperature sensitivity coefficient ($Q_{10AC}$, dimensionless) are set as $6 \times 10^4$ µmol $CO_2$ kg$^{-1}$ Rub s$^{-1}$ and 2.4, respectively. The
compensation point of intercellular $CO_2$ concentrations ($\gamma$, Pa) in the absence of leaf maintenance respiration (MR$_{leaf}$, µmol
$CO_2$ m$^{-2}$ s$^{-1}$) is a function of atmospheric concentration of oxygen gas ($C_{O_2}$, Pa) and the kinetic constants for Rubisco
carboxylation ($K_c$, Pa) and oxygenation ($K_o$, Pa) adapted by temperature (Eq. 13). The $C_{O_2}$ is assumed to be 21% in the
atmosphere (Air, Pa) by volume (Eq. 14). The $K_c$ and $K_o$ values are adapted from their constants at the standard condition





(25 °C, $K_{c25}$ = 40.4 Pa, and $K_{o25}$ = 24800 Pa) to the $T_{air}$ condition, using the $Q_{10}$ values of 2.1 and 1.2, respectively, as the
temperature sensitivity coefficients for variables (Eqs. 15−16). The $MR_{leaf}$ in Eq. 9 is on the PLAI basis (refer to the section of

210    2.1.2.5).

$$A_v = \frac{V_{max}(C_i - \gamma)}{C_i + K_c(1 + \frac{C_{O_2}}{K_o})} - MR_{leaf} \qquad \text{Eq. 9}$$

$$V_{max} = WPN_{leaf}f_{NR}AC \qquad \text{Eq. 10}$$

$$N_{leaf} = \frac{1}{r_{cnl}SLA} \qquad \text{Eq. 11}$$

$$AC = \begin{cases} \dfrac{1.8AC_{25}Q_{10AC}^{(\frac{T_{air}-15}{10})}}{Q_{10AC}} & T_{air} \leq 15\ °C \\ AC_{25}Q_{10AC}^{(\frac{T_{air}-25}{10})} & T_{air} > 15\ °C \end{cases} \qquad \text{Eq. 12}$$

$$\gamma = 0.5K_c \frac{C_{O_2}}{K_0} \qquad \text{Eq. 13}$$

$$C_{O_2} = 0.21Air \qquad \text{Eq. 14}$$

$$K_c = \begin{cases} \dfrac{1.8K_{c25}Q_{10K_c}^{(\frac{T_{air}-15}{10})}}{Q_{10K_c}} & T_{air} \leq 15\ °C \\ K_{c25}Q_{10K_c}^{(\frac{T_{air}-25}{10})} & T_{air} > 15\ °C \end{cases} \qquad \text{Eq. 15}$$

$$K_o = K_{o25}Q_{10K_o}^{(\frac{T_{air}-25}{10})} \qquad \text{Eq. 16}$$

The $A_j$ is simulated by following Eqs. 17−20. Using the Eq. 18 (de Pury and Farquhar, 1997), the rate of electron
transport rate per unit leaf area ($J$, μmol $CO_2$ m$^{-2}$ s$^{-1}$) is simulated as a function of potentially maximum rate of electron
transport rate per unit leaf area ($J_{max}$, μmol $CO_2$ m$^{-2}$ s$^{-1}$), curvature of leaf response of electron transport to irradiance ($\theta_1 = 0.7$)
and PAR effectively absorbed by photosynthesis per unit leaf area ($I_e$, μmol $CO_2$ m$^{-2}$ s$^{-1}$). The $I_e$ is calculated by Eq. 19, based
on total absorbed PAR per unit leaf area ($I$, μmol $CO_2$ m$^{-2}$ s$^{-1}$), spectral correction factor ($f_{sc} = 0.15$) and photons absorbed by
photosynthesis per transported electron (ppe, mol mol$^{-1}$). The $I$ is a function of APAR and PLAI using the unit conversion
coefficient (EPAR) of 4.55 from W m$^{-2}$ to μmol $CO_2$ m$^{-2}$ s$^{-1}$ for the sun and shade leaves (Eq. 20). The values of ppe are 2.6
and 3.5 for C$_3$ and C$_4$ plant, respectively.



$$A_j = \frac{J(C_i - \gamma)}{4.5C_i + 10.5\gamma} - MR_{leaf} \qquad \text{Eq. 17}$$

$$\theta_1 J^2 - (I_e + J_{max})J + I_e J_{max} = 0 \qquad \text{Eq. 18}$$

$$I_e = I\frac{(1 - f_{sc})}{ppe} \qquad \text{Eq. 19}$$

$$I = EPAR\frac{APAR_{sun/shade}}{PLAI_{sun/shade}} \qquad \text{Eq. 20}$$

Equations 6 and 9 are solved for $C_i$, which in turn is introduced into Eq. 17 to solve $A_j$. The smaller value between the $A_v$
and $A_j$ solutions is accepted as the rate of photosynthesis on the PLAI basis.
**2.1.2.3 Litter decomposition**
The carbon and nitrogen from dead leaves and fine roots are directly moved into four litter compartments according to the
specified proportions, while the nitrogen removed from the leaves before senescence is re-translocated. The turnover of live to
dead stems and coarse roots happens daily at a rate determined annually using the annual maximum live woody mass and the
specified live wood turnover rate. The dead stems and coarse roots are received by the coarse woody debris pool which is
fragmented and allocated to the litter pools. The litter pools then decompose and enter into the soil organic matter pools. The
decomposition of shielded cellulose, liable carbon, unshielded cellulose and lignin to unshielded cellulose, liable microbe,
liable humads and resistant humads, as well as the heterotrophic respiration and nitrogen immobilization by microbes during
these processes, are considered in the model, with the actual decomposition being scaled depending on the competing plant
nitrogen demand during allocation. Following Eq. 21 (White $et\ al.$, 2000), the potential nitrogen immobilization ($N_{immo\_p}$, kg N
m$^{-2}$ (3h)$^{-1}$) by microbial decomposition is calculated using the potential decomposed carbon ($C_{decom\_p}$, kg C m$^{-2}$ (3h)$^{-1}$),
fractions of heterotrophic respiration ($f_{HR}$, dimensionless) and C:N ratio of litter ($R_{litCN}$, dimensionless) and soil organic matter
($R_{soilCN}$, dimensionless). The fractions related to the decomposition processes of leaf, fine root, stem and litters, as well as the
maximum rate constants and biomass loss through heterotrophic respiration, are all defined as constants (online supplementary
material of Table S1 and S3). The decomposition rates of litters are also adjusted by the soil temperature (t_adjust) and
moisture (w_adjust) as showed in Eq. 22−23, with the calculated soil water pressure under saturation and the minimum soil
water pressure (Min$_{pressure}$) set as −10 Mpa.





$$N_{\text{immo\_p}} = C_{\text{decom\_p}} \frac{1 - f_{\text{HR}} - \frac{R_{\text{litCN}}}{R_{\text{soilCN}}}}{R_{\text{soilCN}}} \qquad\qquad \text{Eq. 21}$$

$$\text{t\_adjust} = \begin{cases} e^{4.344692 - \frac{1}{(T_{\text{air}} + 273.15) - 227.13}} & T_{\text{air}} \geq -10 \, °C \\ 0 & T_{\text{air}} < -10 \, °C \end{cases} \qquad \text{Eq. 22}$$

$$\text{w\_adjust} = \begin{cases} \dfrac{\ln(\frac{\text{Min}_{\text{pressure}}}{\text{SM}_{\text{pressure}}})}{\ln(\frac{\text{Min}_{\text{pressure}}}{\text{Sat}_{\text{pressure}}})} & \text{SM}_{\text{presssure}} > \text{Sat}_{\text{pressure}} \\ 1.0 & \text{SM}_{\text{presssure}} > \text{Sat}_{\text{pressure}} \\ 0.0 & \text{SM}_{\text{pressure}} < \text{Min}_{\text{pressure}} \end{cases} \qquad \text{Eq. 23}$$

**2.1.2.4 Allocation**

The assimilated carbon allocation and the nitrogen competition between plant nitrogen uptake and litter decomposition are calculated from the potential carbon quantity assimilated ($C_{\text{assi\_p}}$, kg C m$^{-2}$ (3h)$^{-1}$) in photosynthesis and the potential microbial nitrogen demand ($N_{\text{immo\_p}}$, kg N m$^{-2}$ 3h$^{-1}$) in organic matter decay.

The assimilated carbon ($C_{\text{assi\_p}}$, kg C m$^{-2}$ (3h)$^{-1}$) available to allocate is the difference between gross primary productivity (GPP, kg C m$^{-2}$ (3h)$^{-1}$) and MR (kg C m$^{-2}$ (3h)$^{-1}$) of all live tissues. If the difference is negative, it means a carbon pool deficit; and thus the available carbon for allocation is first allocated to alleviate the deficit. All new allocations to other organs or tissues are constrained by the new leaf carbon allocation (Waring and Running, 2007). The carbon required for per unit of leaf growth ($C_{\text{allometry}}$, dimensionless), which is the carbon allometry, is calculated based on the fractions of user-defined allocation rates and growth respiration ($f_{\text{GR}}$, dimensionless), which in turn accounts for 30% of the total carbon in new tissue (Eq. 24). The key allocation ratios include new fine root carbon to new leaf carbon ($r_{\text{frtol}}$, dimensionless), new stem carbon to new leaf carbon ($r_{\text{stol}}$, dimensionless), new live wood carbon to new total wood carbon ($r_{\text{lwtow}}$, dimensionless) and new coarse root carbon to new stem carbon ($r_{\text{crtos}}$, dimensionless). According to Eq. 25, the associated nitrogen needed by per unit of leaf growth ($N_{\text{allometry}}$, dimensionless), which is the nitrogen allometry, is calculated based on carbon allometry and the C:N ratios of leaf ($R_{\text{lCN}}$, dimensionless), fine root ($R_{\text{frCN}}$, dimensionless), live wood ($R_{\text{lwCN}}$, dimensionless) and dead wood ($R_{\text{dwCN}}$, dimensionless) (Eq. 25). The nitrogen demand for plant growth ($N_{\text{assi\_p}}$, kg N m$^{-2}$ (3h)$^{-1}$) is predicted by the potential





assimilated carbon ($C_{\text{assi\_p}}$, kg C m$^{-2}$ (3h)$^{-1}$), carbon allometry ($C_{\text{allometry}}$, dimensionless) and nitrogen allometry ($N_{\text{allometry}}$, dimensionless), using Eq. 26. The total nitrogen demand ($N_{\text{demand}}$, kg N m$^{-2}$ (3h)$^{-1}$) is the sum of the potential nitrogen immobilization ($N_{\text{immo\_p}}$, kg N m$^{-2}$ (3h)$^{-1}$) and the nitrogen demand for plant growth ($N_{\text{assi\_p}}$, kg N m$^{-2}$ (3h)$^{-1}$).

$$C_{\text{allometry}} = (1 + f_{\text{GR}})[1 + r_{\text{frtol}} + r_{\text{stol}}(1 + r_{\text{crtos}})] \qquad \text{Eq. 24}$$

$$N_{\text{allometry}} = \frac{1}{R_{\text{lCN}}} + \frac{r_{\text{frtol}}}{R_{\text{frCN}}} + \frac{r_{\text{stol}}r_{\text{lwtow}}(1 + r_{\text{crtos}})}{R_{\text{lwCN}}} + \frac{r_{\text{stol}}(1 - r_{\text{lwtow}})(1 + r_{\text{crtos}})}{R_{\text{dwCN}}} \qquad \text{Eq. 25}$$

$$N_{\text{assi\_p}} = C_{\text{assi\_p}} \frac{N_{\text{allometry}}}{C_{\text{allometry}}} \qquad \text{Eq. 26}$$

If the soil available mineral nitrogen ($N_{\text{soil}}$, kg N m$^{-2}$ (3h)$^{-1}$) can satisfy the total nitrogen demand, the actual allocation and microbial decomposition are equal to the potential values. The plant nitrogen demand is first addressed by the retranslocated nitrogen and then the soil mineral nitrogen. The nitrogen demand for decomposition is totally from the soil mineral nitrogen. Otherwise, the potential microbial decomposition is adjusted by a dimensionless factor of FPI, which is defined with $N_{\text{soil}}$ and the ratio $N_{\text{immo\_p}}$ to $N_{\text{demand}}$ (Eq. 27). The $N_{\text{demand}}$ is still provided by the retranslocated nitrogen and soil mineral nitrogen pools. If the $N_{\text{demand}}$ can be satisfied by the nitrogen pools, the actual nitrogen allocations are equal to the potential nitrogen allocations. Otherwise, the actual nitrogen allocations are proportionally reduced (Wang *et al.*, 2009).

$$\text{FPI} = \frac{N_{\text{soil}} \dfrac{N_{\text{immo\_p}}}{N_{\text{demand}}}}{N_{\text{immo\_p}}} \qquad \text{Eq. 27}$$

**2.1.2.5 Respiration**

The maintenance respiration (MR, kg C m$^{-2}$ (3h)$^{-1}$) is a function of the nitrogen concentration of living plant tissues ($N$, kg N m$^{-2}$), temperature sensitivity ($Q_{10} = 2.0$) and $T_{\text{air}}$ (Eq. 28). The plant nitrogen concentration is assumed to linearly affect MR, with a relationship of 0.218 kg C d$^{-1}$ kg$^{-1}$ N (Ryan, 1991). The MR is separately calculated for leaves, fine roots, live stems and live coarse roots. The leaf maintenance respiration (MR$_{\text{leaf}}$, μmol CO$_2$ m$^{-2}$ s$^{-1}$) on the PLAI basis is separately calculated for sun and shade leaves, based on the nitrogen concentration on the PLAI basis ($N_{\text{leaf}}$, kg N m$^{-2}$), $Q_{10}$, $T_{\text{air}}$ and the molar mass of carbon ($M_{\text{CO}_2} = 12$ g mol$^{-1}$) (Eq. 29).





$$\mathrm{MR} = \frac{0.218 N Q_{10}^{\frac{T_{\mathrm{air}}-20}{10}}}{8} \qquad \text{Eq. 28}$$

$$\mathrm{MR_{leaf}} = \frac{0.218 N_{\mathrm{leaf}} Q_{10}^{\frac{T_{\mathrm{air}}-20}{10}}}{86400 \times 10^{-9} M_{\mathrm{CO_2}}} \qquad \text{Eq. 29}$$

The growth respiration (GR, kg C m$^{-2}$ (3h)$^{-1}$) is calculated separately for plant organs of leaves, fine roots, live stems, and
live coarse roots. For a plant organ, its GR is calculated by allocating the carbon in this organ to its GR pool. The stored GR
pool of this organ is fully or partly converted to the growth in the next year using two key parameters. One is the fraction of
carbon respired for growth ($g_1 = 0.3$). And the other is the proportion of growth respiration to release at fixation ($g_2 = 1.0$). The
carbon pools of plant organs are calculated in the allocation module.

**2.1.2.6 Mortality**

The mortality is calculated for all plant organs using user-defined rates and the dead tissues enter the pools of coarse
woody debris and litters at each time step. In addition, plant mortality due to fire can be simulated using a user specified rate,
which results in the losses of carbon and nitrogen to the atmosphere.

**2.2 Description of observed data**

The observed data for model validation were obtained from the Chinese Terrestrial Ecosystem Flux Observation and
Research Network (ChinaFLUX) which was established in 2002 (http://www.chinaflux.org/enn/index.aspx). Using the
open-path eddy covariance techniques, continuous measurements of carbon and water fluxes have been conducted since then
in the major typical forest ecosystems of the eastern Asia, including the temperate site (42°24′09″N, 128°05′45″E, 738 m a.s.l.)
in Changbai Mountain (CBM) with mixed forest of evergreen needle leaf and deciduous broad leaf trees, The subtropical site
(26°44′29″N, 115°03′29″E, 102 m a.s.l.) at Qianyanzhou (QYZ) with artificial evergreen coniferous forest and the subtropical
site (23°10′25″N, 112°32′04″E, 300 m a.s.l.) in the Dinghu Mountain (DHM) with evergreen mixed forests of broad and
needle leaf trees (Yu *et al.*, 2008; Yu *et al.*, 2013; Zhang *et al.*, 2019a). All the forests at the three sites have been treated as
natural reservation area without management. The available datasets for the model calibration and validation at the three sites



included daily GPP, ER, NEE and ET measured in 2003 to 2010. Meanwhile, the observed hourly meteorological datasets,
including air temperature, precipitation, radiation, wind speed and relative humidity, were used as model inputs to drive the
simulation.
The CBM forest site is located in northeast China and situated within the National Natural Conversation Park of the
Changbai Mountains, in the eastern Jilin Province (Guan *et al.*, 2006). The region is subjected to a temperate continental
monsoon climate, with an average annual air temperature of 3.6 °C and annual precipitation of 695 mm (1985−2005) (Yu *et al.*,
2008). A homogeneous mixture of broad leaf deciduous and evergreen coniferous (red pine) forest widely distributes in the
natural reservation area (Wu *et al.*, 2021).
The QYZ site is located in southeast China and situated the Jiangxi Province. The region is subjected to a subtropical
continental monsoon climate, with mean annual air temperature of 17.9°C and annual precipitation of 1475 mm (1985−2007)
(Wen *et al.*, 2010). The artificial pure coniferous forests were established in 1985 (Dai *et al.*, 2021).
The DHM site is located in south China and situated in the Dinghu Mountains Biosphere Reserve of the Guangdong
Province. The region is subjected to a typical subtropical monsoon humid climate, with an average annual air temperature of
22.3 °C and annual precipitation of 1678 mm, of which 80% falls between April and September (Zhou *et al.*, 2013). The carbon
and water fluxes were observed in a mixed forest of evergreen broad leaf and conifer trees, which is a major forest type in
low-latitude subtropical eastern Asia (Li *et al.*, 2012).
More detailed site descriptions can be referred to the online supplementary material of Table S2.
**2.3 Model simulation**
The modified CNMM-DNDC was calibrated and validated using the observed data of carbon and water fluxes from 2003
to 2007 and 2008 to 2010, respectively, in each of the three forest sites. The required input data of climate ($T_{air}$, precipitation
(Prec), wind speed (Wind), solar radiation (Rad), and relative humidity (Hum) in the resolution of 3-hour) and basic soil
properties (soil clay fraction, organic matter content (SOM), total nitrogen, pH and bulk density (BD)) were primarily
obtained from the National Ecosystem Science Data Center (NESDC; http://www.cnern.org.cn). The climate data out of the
observational period were obtained from the China meteorological forcing dataset (1979–2018) (https://data.tpdc.ac.cn).





Additional required soil parameters also as model inputs, including field capacity, wilting point and saturated hydrological
conductivity, were calculated using pedo-transfer functions (Li *et al.*, 2019). The mixed forests were assumed as
homogeneously distributed deciduous broad leaf tree (DBT) and evergreen needle leaf tree (ENT) for the CBM and
evergreen broad leaf tree (EBT) and evergreen needle leaf tree (ENT) for the DHM. The vegetation type at the QYZ site was
a pure ENT. The required parameters for forest simulation, including forest type, carbon contents of leaf and stem and
eco-physiological parameters (Table S3), were primarily adapted from the field observations provided by the NESDC or
from the peer reviewed literatures (Zeng *et al.*, 2008; Li, 2018; Li, 2019). The simulated soil profile (0−1.5 m in depth) was
divided into 16 layers, with layer thicknesses of 0.05, 0.1 and 0.5 cm for the 0–0.5, 0.5–1 and 1–1.5 m depths, respectively.
In order to stabilize the carbon and nitrogen dynamics and thus reduce the residual effects of initial conditions (Zhang *et al.*,
2015), especially for the carbon balance among different pools, which requires spin-up for at least 10 years (Palosuo *et al.*,
2012), the model simulation at each forest site was pre-run for 13 years.
**2.4 Sensitivity analysis**

The one-at-a-time (OAT) sensitivity analysis was adopted to examine the influences of newly involved parameters in

the modified CNMM-DNDC model and the inputs as primary drivers on the simulated GPP and ER from the investigated
forest ecosystems. As the NEE was the difference between ER and GPP, it was not considered in the sensitivity analysis.
The simulations during the validation periods (2008–2010) for the three forest sites were regarded as the baseline. The
eco-physiological parameters required for the newly established forest growth module were involved in the sensitivity
analysis, which are detailed in Table S3. In each OAT sensitivity analysis of parameters, a parameter was altered by $\pm 20\%$,
$\pm 15\%$ and $\pm 10\%$, with the others remaining constant (White *et al.*, 2000). In addition, meteorological variables (i.e., $T_{air}$,
Rad, Wind and Hum as 3-hourly means and Prec as 3-hourly totals during the validation periods) and soil properties (i.e.,
soil clay fraction, pH, SOM content and BD) were involved in the sensitivity analysis of model inputs. In the OAT
sensitivity analysis of model inputs, the values of a meteorological variable during the validation periods and the value of a
soil property were altered by $\pm 20\%$, $\pm 15\%$ and $\pm 10\%$, but with exception for $T_{air}$, BD and pH; while the other
meteorological inputs remained unaltered and the other soil properties remained constant. The 3-hourly average of $T_{air}$ during





the validation period was altered within the range of –3 to +3 $^{\circ}$C with an interval of 1 $^{\circ}$C. Considering that real soil BD and

pH usually vary with narrow amplitudes, the former property was altered within the ranges of 0.97 to 1.17 (CBM), 1.20 to

1.40 (QYZ) and 1.28 to 1.48 (DHM), with an interval of 0.04, and the latter within the ranges of 4.64 to 5.64 (CBM), 4.1 to

4.9 (QYZ) and 3.3 to 4.3 (DHM), with an interval of 0.2, respectively. The sensitivity of annual GPP and ER fluxes to an

examined parameter or input variables was evaluated using a sensitivity index ($I$, %) following Majkowski *et al.* (1981).

Each $I$ value was calculated with Eq. 30.

$$I = \frac{100 \sum_{i=1}^{n} (|M_i - M_{\text{baseline}}|)}{n M_{\text{baseline}}} \qquad \text{Eq. 30}$$

In Eq. 30, $M_i$ and $M_{\text{baseline}}$ denote the simulated annul GPP or ER corresponding to the $i^{\text{th}}$ altered value(s) and the

baseline of the examined parameter or input variables, respectively; and $n$ represents the total number of alternation for a

parameter or input variable, which was set as 6 in this study. The three-year mean and standard deviation of a forest type at

each site were reported for the value of index ($I$), indicating the sensitivity of annual GPP or ER fluxes to a parameter or an

input variable.

**2.5 Statistics and analysis**

The statistical criteria applied for evaluating the model performance in this study (Eqs. 31−33) included (i) the

normalized root mean square error (NRMSE), (ii) the Nash–Sutcliffe efficiency (NSE), and (iii) the determination coefficient

($R^2$) and slope of a significant linear zero-intercept regression (ZIR) (Nash and Sutcliffe, 1970; Willmott and Matsuurra, 2005;

Moriasi *et al.*, 2007; Congreves *et al.*, 2016; Dubache *et al.*, 2019). A value of NRMSE closer to 0 indicated a better simulation.

The NSE value (ranging from minus infinity to 1) is defined to compare the overall deviation of the simulations from the

observations with the observed variance (Nash and Sutcliffe, 1970). An NSE of 1 indicates the best simulation, and a value

between 0 and 1 shows smaller overall deviation than the observed variance and thus an acceptable model performance, but

otherwise worse. A better model performance is also indicated by a slope and an $R^2$ value simultaneously closer to 1 in a

significant zero-intercept linear regression of observations against simulations (Dubache *et al.*, 2019). The absolute

divergence of the slope from 1 represents the average bias of the simulations from the observations (Zhang *et al.*, 2019b).





$$\text{NRMSE} = \frac{1}{|\bar{O}|}\sqrt{\frac{\sum_{i=1}^{n}(S_i - O_i)^2}{n}} \qquad\qquad \text{Eq. 31}$$

$$\text{NSE} = 1 - \frac{\sum_{i=1}^{n}(S_i - O_i)^2}{\sum_{i=1}^{n}(O_i - \bar{O})^2} \qquad\qquad \text{Eq. 32}$$

$$R^2 = 1 - \frac{\sum_{i=1}^{n}(O_i - \hat{O}_i)^2}{\sum_{i=1}^{n}(O_i - \bar{O})^2} \qquad\qquad \text{Eq. 33}$$

In Eqs. 31−33, $i$ and $n$ ($i = 1, 2, …, n$) denote the $i^{th}$ pair and the total number of pairs of the model simulated ($S$),
observed ($O$) and predicted ($\hat{O}$) by linear regression, respectively, with the $n$ values given in Tables 1−2 and $\bar{O}$ denoting the
mean value of $n$ observations.
In addition, the Taylor diagram was also used to evaluate the performances of the original and modified model versions.
A Taylor diagram for all the three field sites was drawn for one of the four examined variables, i.e., GPP, ER, NEE and ET.
It was used to simultaneously demonstrate the correlation coefficients ($r$) between the simulations and observations of daily
fluxes, the centred root mean square difference (RMSD) and the standard deviations (SD) of the observations (the horizontal
axis) and the model simulations. A normalized SD (NSD) of observations was 1 and the corresponding NSD of simulations
was given as the ratio of the standard deviation of simulations to that of observations. A Taylor diagram would indicate a
better model performance with $r$ closer to 1 and a distance closer to zero between the RMSD of simulations and the NSD of
observations.
The SPSS Statistics Client 19.0 (SPSS Inc., Chicago, USA), the Origin 8.0 (OriginLab, Northampton, MA, USA) and the
R software packages were applied for the statistical analysis and graphical comparison. The source code and executive
program of the improved model, as well as the input data can be obtained from Zhang *et al.* (2024).
**3 Results**
**3.1 Model performances in simulating carbon and water fluxes**
The daily GPP observed at the three forest sites all showed significant seasonal variation trends, which were consistent
with the dynamics of air temperature. However, the seasonal patterns at the DHM site that is located in the south margin of
subtropical region were much weaker than those at the other two sites. For the daily GPP simulations, as compared to the
observations, the original model generally resulted in much earlier appearance of the seasonal peak at the CBM site (Fig. 1a),





larger inter-day fluctuations at the QYZ site and larger seasonal peak fluxes at the CBM and DHM sites (Figs. 1a, 2a and 3a).
The modified model overcome these poor performances of the original model, with the NRMSE, NSE and $R^2$ and slope
values of 0.33−0.41, 0.08−0.89, 0.18−0.90, and 0.85−0.94 respectively (Figs. 1a, 2a and 3a; Table 1). At each site, the
modified model generally performed better than the original model, as showed in the Taylor diagram (Fig. 4a). Among the
three sites, the modified model showed the best performance at the CBM site, with NSE (~0.88), $R^2$ (~0.89) and slope
(~0.94), all beings close to 1. In comparison, especially at the CBM and DHM sites, the daily GPP fluxes simulated by the
original model were much higher than the observations during the growing season, which led to the unacceptable statistics
(Table 1). The observed annual GPP fluxes during the eight years ranged from 13.3 to 16.6, 16.5 to 19.3 and 12.5 to 15.5 Mg
C ha$^{-1}$ yr$^{-1}$ at the CBM, QYZ and DHM, respectively. The corresponding annual simulations by the original versus modified
models totalled 12.4−18.5 versus 14.5−18.3, 14.4−19.2 versus 17.6−20.3, and 12.8−14.0 versus 12.7−15.2 Mg C ha$^{-1}$ yr$^{-1}$,
respectively (Fig. 5a−c), with the modified model resulted in an NSE over 0.60. Thus, the modified model showed better
performances in simulating the annual GPP fluxes than the original model (Table 2). The validation results of GPP showed
that the modified model not only performed much better in simulating the daily dynamics during multiple years, but also
more effectively captured the inter-annual variations.
The daily observed ER showed similar dynamics with those of GPP, but the peak values at the CBM sites were nearly
double of those at the DHM site. For the daily ER fluxes, the original model resulted in earlier appearance of the seasonal
peaks at the CBM site, overestimated seasonal fluxes at the CBM and DHM sites and larger inter-day fluctuations or even
underestimation in summer at the QYZ site. In comparison, the modified model could successfully overcome these poor
performances (Figs. 1b, 2b and 3b). It could generally well capture the seasonal patterns, with much smaller deviations,
reporting the NRMSE, NSE, $R^2$ and slope values of 0.18−0.27, 0.62−0.93, 0.68−0.93 and 0.92−1.05, respectively, during
both the calibration and validation periods (Table 1). In contrast, the original model performed poorly in simulating the daily
ER fluxes of the three sites during both periods, showing NRMSE, NSE and $R^2$ ranged from 0.51 to 0.94, −1.32 to 0.25, 0.11
to 0.40 and 0.45 to 0.83, respectively (Table 1). The Taylor diagram also indicates the obvious improvements of ER
simulations by the modified model for the three forest sites, with significant higher correlation coefficients (Fig. 4b). For the
annual ER fluxes at the CBM, QYZ and DHM sites, the observations varied from 11.3 to 14.7, 12.1 to 15.7 and 9.0 to 11.4





Mg C ha$^{-1}$ yr$^{-1}$, respectively. The corresponding simulations by the original versus modified models totalled 10.8−14.9
versus 11.1−16.2, 9.9−13.0 versus 12.7−14.3, and 9.5−10.3 versus 9.5−11.1 Mg C ha$^{-1}$ yr$^{-1}$, respectively (Fig. 5d−f). The
simulations by the modified model were much better than the original model, showing NSE values of 0.56 versus 0.05 for all
cases (Table 2). These results on the model performances indicate that the modified model version developed in this study is
capable of well simulating the sum of $CO_2$ effluxes due to autotrophic respiration and soil heterotrophic respiration in
different forest types subject to subtropical to temperate monsoon climates in east Asia.
The observed daily NEE showed significant seasonal patterns at the CBM site, while less obvious seasonal patterns at
the other two sites (Figs. 1c, 2c and 3c). The Taylor diagram (Fig. 4c) illustrates that the modified model better simulated the
daily NEE fluxes of the CBM during either the calibration or validation period but not at the QYZ and DHM sites (Table 1),
as compared to the original model. However, the corresponding statistics for the NEE validation, with values of 1.05 to 2.85,
−0.11 to 0.47, 0.02 to 0.61 and 0.15−0.41 for the NRMSE, NSE, $R^2$ and slope, respectively, were generally worsen than
those for the GPP and ER (Table 1). The simulations by the original model did not match with the observations due to the
deviations in capturing the seasonal dynamics of GPP and ER. At the CBM site, especially, the original model resulted in
unreasonably intensive negative NEE in the spring season while the modified model in generally performed pretty good for
the dynamical daily NEE fluxes of this site (Fig. 1c). At the QYZ site, both model versions showed comparable
performances in simulating the daily NEE fluxes, though the modified model obviously improved its performances in
simulating the dynamical daily fluxes of both GPP and ER (Fig. 2c). At the DHM site, the simulations by both model
versions showed significant disagreement with the measured daily NEE fluxes in the winter season and occasionally also in
the summer season, though the modified model greatly improved the simulations of daily GPP and ER fluxes (Fig. 3c). The
annually accumulated NEE fluxes observed at the CBM, QYZ and DHM sites in 2003−2010 ranged from −3.8 to −0.7, −5.5
to −3.6 and −5.2 to −2.1 Mg C ha$^{-1}$ yr$^{-1}$, respectively. The annual NEE fluxes simulated by the original versus modified
models ranged from −3.6 to −1.4 versus −4.2 to −1.8, −6.1 to −4.6 versus −6.8 to −3.9, and −3.0 to −3.7 versus −4.1 to −3.3
Mg C ha$^{-1}$ yr$^{-1}$ at the CBM, QYZ, and DHM sites, respectively (Fig. 5g−i). Despite the worse simulations of GPP and ER by
the original model, the simulated annual NEE fluxes were comparable between the two model versions (Table 2).





Overall, for the carbon fluxes of the three forest sites during the multiple-year periods, the modified model effectively captured the dynamical GPP and ER, followed by the NEE (Fig. 1−3). For the different forest types, the modified model better predict the annual NEE of the pure ENT (QYZ) and the mixing forest of ENT and DBT (CBM) than the mixing forest of ENT and EBT (DHM) (Table 2). Following the definition of NEE (i.e. NEE = ER – GPP), the NRMSE of simulated daily and annual NEEs could be propagated from those of GPP and ER simulations. For the daily NEEs during the validation period, the propagated NRMSE values were 0.49, 0.40 and 0.42 at the CBM, QZY and DHM, respectively (0.44 on average across the three sites). For the annual NEEs, the propagated NRMSE value across all the three sites during the three validation years was 0.13. These propagated NRMSEs for the modified model simulations, as Table 1 and 2 list, were much smaller than the NRMSEs of direct NEE simulations (0.44 versus 1.92 on average for the daily fluxes and 0.13 versus 0.51 for the annual fluxes), or as compared to the NRMSEs propagated from those of the original model simulations on ER and GPP (0.44 versus 1.02 on average and 0.13 versus 0.23 for the daily and annual NEE fluxes, respectively).

The seasonal dynamics of daily ET fluxes were consistent with those of daily GPP due to the contribution of transpiration which is closely related with plant growth. At the CBM sites, both model versions showed comparable good performance in simulating the dynamical ET fluxes. At the other two sites, the original model failed to capture the seasonal dynamics of ET fluxes, but the modified model performed as good as it did at the CBM site (Figs, 1d, 2d and 3d). The modified model showed NRMSE, NSE, $R^2$ and slope values of 0.32−0.46, 0.59−0.81, 0.49−0.82 and 0.61−0.77 for simulating the daily ET fluxes during the three-year validation period, respectively, especially with great improvement of NSE (0.80 versus 0.00 at QYZ, and 0.59 versus −0.55 at DHM) as compared to the original model (Table 1). The Taylor diagram also suggested that ET simulations by the modified model were significantly better at the sites of QYZ and DHM, as compared to the original model (Fig. 4d). The observed annual ET quantities in 2000−2008 varied from 392 to 495, 510 to 768 and 574 to 720 mm at the CBM, QYZ and DHM sites, respectively. The annual ET simulated by the original versus modified model versions amounted 332−485 versus 394−522, 696−800 versus 654−732, and 583−726 versus 614−725 mm at the CBM, QYZ and DHM, respectively (Fig. 5j−l). In comparison, the modified model better captured the inter-annual variations of ET, with an NSE value greater than 0.60 (Table 2). In general, the modified model with the updated plant growth module performed better in not only in simulating the daily ET dynamics, but also the inter-annual variations.





In comparison with the original model, according to the statistics of daily validation fluxes for all the examined
variables including ($n = 12$) and excluding ($n = 9$) the NEE of all the three sites (Table 1), the modified model on average
reduced the NRMSE by 38% ($p < 0.01$) and 50% ($p < 0.01$), increased the NSE from −0.26 to 0.54 and −0.34 to 0.68,
enlarged the $R^2$ by 85% (from 0.39 to 0.72, $p < 0.05$) and 90% (from 0.42 to 0.80, $p < 0.05$), and reduced the absolutely
deviations of the slopes from 1 (from 0.35 to 0.12, $p < 0.01$ and from 0.33 to 0.04, $p < 0.001$), respectively. These results
further showed the obviously improved performances of the modified model in simulating daily fluxes of GPP, ER and ET,
particularly reduced significantly the biases of daily GPP and ER.
For the statistics of NRMSE, NSE, slope and $R^2$ on the annual GPP and ER at the three sites in the validation years, the
modified model showed slightly significant improvement ($n = 8$, $p < 0.01$), while those on the annual NEE and ET were
comparable between the modified and original model (Table 2).

**3.2 Sensitivity of carbon fluxes to examined parameters or factors**

For the newly incorporated eco-physiological parameters, as Figs 6−7 and Figs S1−2 illustrate, the modified model
simulations of both annually cumulated GPP and ER fluxes at the three sites were sensitive to canopy average specific leaf
area (SLA, p30), fraction of leaf nitrogen in Rubisco (FLNR, p32), annual fraction of leaf and fine root turnover (LFRT, p6)
and maximum stomatal conductance ($g_{CO_{2max}}$, p33). FLNR, LFRT and SLA were the parameter with the highest sensitivity
for the mixed forests at the CBM, QYZ and DHM sites, respectively. Both annual GPP and ER fluxes of the mixed forests at
the CBM and DHM sites were more sensitive to the parameter of FLNR than those of ENT at the QYZ site. Compared with
the forests in the temperate region, the annual GPP and Re fluxes of subtropical forests were more sensitive to the parameter
of LFRT. The values of sensitivity index were increased with the decreased latitude. For annually cumulated GPP fluxes,
SLA was the parameter with higher sensitivity for the ENT in different climate zones, while FLNR was the most sensitive
parameter for the DBT in the temperate region and the EBT in the subtropical region (Figs S1). The sensitivity index of
FLNR for the DBT was nearly twice of that for the ENT at the CBM site, and thus FLNR was identified as the most
sensitive parameter for the annual GPP fluxes at the CBM site. Similarly, the relatively high index of LFRT for the ENT at
the DHM site led to intensive responses of the simulated annual GPP fluxes to the changes of LFRT. With regard to the





sensitive parameters of annual ER fluxes, there were SLA, fine roots (C:N$_{fr}$, p17), LFRT and $g_{smax}$ for the ENT and, FLNR,
LFRT and carbon to nitrogen ratios in leaves (C:N$_{leaf}$, p15) for the DBT and EBT (Figs S2).
For the meteorological variables, Figs. 6−7 demonstrate that the simulated annual GPP and ER fluxes were sensitive to
solar radiation and air temperature at the CBM and DHM sites with mixed forests, while were solar radiation and humidity at
the QYZ site. The values of sensitivity index were comparable among three sites. For the ENT, the annual GPP fluxes were
most sensitive to the solar radiation and humidity at the three sites in different climate zones. For the DBT and EBT, the
annual GPP and ER fluxes were most sensitive to the solar radiation and air temperature (Figs. S1−2). Meanwhile, both the
annual GPP and ER fluxes of the different forests were not sensitive to the examined soil properties of clay fraction, soil
organic matter, pH and bulk density with sensitivity indexes less than 1%.
**4 Discussions**
**4.1 Model simulations on carbon fluxes**
As GPP is an essential flux component in the carbon budget of forests at site to global scales, accurate estimation of
forest GPP is urgently required to understand and assess the dynamics of global carbon cycle, predict future trends and
ensure long term security of the ecosystem services (Campbell *et al.*, 2017; Cook-Patton *et al.*, 2020). Light use efficiency
(LUE) models, such as CASA, MODIS, can adequately simulate the spatial and temporal dynamics of GPP due to the usage
of extensive satellite observations (Potter *et al.*, 1993; Running *et al.*, 2004). By comparing and assessing major algorithms
and performances of seven LUE models, Yuan *et al.* (2014) have concluded that most models can effectively capture the
temporal variations and magnitudes of daily GPP in deciduous broad leaf forests and mixed forests (with $R^2$ of 0.6 to 0.8).
However, this type of model cannot be applied for prediction due to the dependence on satellite data that are only available
historically (Yuan *et al.*, 2014). As Raczka *et al.* (2013) have indicated, reproducing the inter-annual variability of GPP is a
challenge for both LUE and terrestrial biosphere models. Process-oriented models are designed to enable prediction of future
carbon cycle through describing the physical and mechanistic processes occurring over time (Srinet *et al.*, 2023). The
comparison studies of model simulation against the observations in North American (Keenan *et al.*, 2012; Raczka *et al.*,
2013) have showed that, as compared to LUE models, process-oriented models possess some skills in simulating the





inter-annual variations (with $R$ of 0.09 to 0.46). In the present study, the modified process-oriented model well simulated the
multiple-year dynamics of daily GPP fluxes (with $R^2$ of 0.5 to 0.9) and showed acceptable performance in simulating
inter-annual GPP variations (with $R^2$ of 0.8). These results are comparable with or even better than the simulations of the
Biome-BGC model for the three forest sites involved in our study (Wen, 2019; Fan, 2021). The worst performance for the
forest at the DHM site may be attributed to the errors in simulating the photosynthesis of EBT due to the difficulty in
modelling the subtle changes in the leaf phenology. Such a difficulty has also been encountered by previous studies (Raczka
*et al.*, 2013; Yuan *et al.*, 2014). In addition, previous study has showed that assimilating satellite data, e.g., LAI, can
significantly improve the performance of process-oriented models in simulating daily GPP (Yan *et al.*, 2016). Therefore, for
the large scale studies, assimilating satellite data may provide a solution to further improve the ability of the modified
CNMM-DNDC model for simulating the spatial and temporal dynamics of forest GPP.

An ER flux consists of the flux components of plant autotrophic respiration and soil heterotrophic respiration, which is

derived from the organic carbon of all organisms in an ecosystem (Chapin et al., 2012). The uncertainties of ER simulations
can lead to the bias of other variables in carbon cycle, such as NEE and net primary production (Fang *et al.*, 2022). Lu *et al.*,
(2021) have found that terrestrial ecosystem models result in large divergences in simulating annual soil respiration and its
components (with $R^2$ less than 0.5). In this study, our modified model well simulated the multiple-year ER at the three sites,
with $R^2$ around 0.6 which was comparable with simulations based on the random forest model (Lu *et al.*, 2021). In addition,
compared with the ER at the DHM site, the intensive fluxes in the growing season at the CBM were primarily attributed to
the latitudinal gradients of temperature and precipitation along the North-South Transect in Eastern China (Yu *et al.*, 2008).
Field experiments observed highly spatial variability in $Q_{10}$ as a temperature sensitivity parameter of ER, which is usually a
key parameter in the process-oriented models (Yu *et al.*, 2008; Anav *et al.*, 2013). However, inadequately considering the
spatial heterogeneity of $Q_{10}$ in process-oriented models may lead to divergences in simulating ER. Therefore, mechanistic
parameterization of $Q_{10}$ to better reflect the spatial heterogeneity of this parameter is still needed for large scale applications
of our modified model to estimate the carbon budgets of terrestrial ecosystems.

As compared to GPP and ER, the modified model performed worse in simulating NEE at either the daily or annual

scale. The highest uncertainty in predicting NEE is consistent with other studies, which may be due to the compounding





effects of GPP and ER errors (Fang *et al.*, 2022). Using the default and calibrated parameters or the assimilated satellite LAI
data, Liu (2018) have evaluated the performances of the Biome-BGC model in simulating the daily NEE of three years at the
three forest sites involved in our study and resulted in RMSEs of 9.3 to 31.6 kg C ha$^{-1}$ d$^{-1}$. Our results during the validation
periods ranged from 14.6 to 16.9 kg C ha$^{-1}$ d$^{-1}$, which were comparable to the results using the calibrated parameters in the
above study. At the annual scale, the correlation between simulated and observed GPP is consistently higher than that of
NEE, indicating the high sensitivity of NEE to small relative errors in large GPP fluxes (Raczka *et al.*, 2013). Meanwhile,
the correlation of ER between simulations and observations was lower than that of GPP, suggesting that ER may be the main
contributor to the poor simulation of inter-annual variability in NEE. Although process-oriented models can effectively
simulate the different types of carbon fluxes by incorporating the inter-annual influences of temperature and soil moisture,
the simulated NEE in this study, as well as others (Keenan *et al.*, 2012; Raczka *et al.*, 2013), can draw a conclusion that
process-oriented models do not adequately explain the observed inter-annual variability in NEE, yet. In addition, despite the
much better performances of our modified model in simulating both the daily and annual GPP and ER, the statistics of
simulated NEE were almost the same between the original and the modified model versions. These results suggest the
necessity of validating each component of carbon fluxes from ecosystems so as to avoid the offsetting of model errors.
**4.2 Sensitivity analysis**

As plant autotrophic respiration consumes nearly 50% of the carbon available from photosynthesis, ER is closely tied to

plant productivity, and thus the strong positive correlation between GPP and ER is presented in models (Ryan, 1991; Piao *et*
*al.*, 2013; Jagermeyr *et al.*, 2014). The analysis results of this study also indicated that the sensitive parameters of ER were
generally consistent with those of GPP. The parameters that were most sensitive for GPP and ER were those controlling the
$CO_2$ assimilation rate in photosynthesis and influencing respiration, such as SLA, FLNR, LFNR, $g_{CO_{2max}}$, C:N$_{leaf}$ and C:N$_{fr}$.
This result is consistent with those reported by White *et al.* (2000), Raj *et al.* (2014), Miyauchi *et al.* (2019), Ren *et al.* (2022)
and Srinet *et al.* (2023). Specific leaf area was identified as a sensitive parameter for both GPP and ER at the three sites as
the ENT were sensitive to SLA at all sites (Fig. S1). This parameter is widely used in process-oriented models for calculating
LAI which influences all aspects of canopy physiology (White *et al.*, 2000). Miyauchi *et al.* (2019) have concluded that SLA





strongly affects aboveground woody and leaf carbon density in *Eucommia ulmoides* plantations on the Loess Plateau.
Therefore, it is not surprising that the variations in SLA result in significant corresponding changes in GPP, as well as ER
(Srinet *et al.*, 2023). Our results also showed that both GPP and ER were sensitive to the parameter of FLNR, especially for
the broad leaf trees. Carboxylation is the first step of atmospheric $CO_2$ assimilation, using the catalyst of Rubisco which is
probably the most abundant protein on earth (Stitt and Schulze, 1994; White *et al.*, 2000). The fraction of leaf nitrogen in
Rubisco controls the potential rate of carboxylation ($V_{max}$). This potential rate is a significant state variable with high spatial
variability that is difficult to obtain (Houborg *et al.*, 2012). Thus, the intensive impacts of FLNR on GPP and ER are
reasonable (Raj *et al.*, 2014). In addition, LFRT was identified as a sensitive parameter for the forests in the subtropical
regions. For deciduous trees, all the leaf and fine root carbon pools are fully turned over every year and thus LFRT is set as
1.0. For evergreen trees, however, LFRT is given as the inverse of mean leaf longevity (White *et al.*, 2000). Net
photosynthetic capacity has been found to be negatively related with leaf longevity (Reich *et al.*, 1999). It is determined by
the positive relationship between LFRT and FLNR in our model. Due to the regulating effects of stomatal conductance on
water loss and carbon assimilation (White *et al.*, 2000; Miyauchi *et al.*, 2019), $g_{CO_{2max}}$ was recognized in our study as a
relatively sensitive parameter. In process-oriented models, e.g., the modified CNMM-DNDC, C:$N_{leaf}$ determines the nitrogen
required for leaf construction and the content of chlorophyll, thereby affecting photosynthesis, leaf respiration and net
primary cumulative productivity (White *et al.*, 2000; Ren *et al.*, 2022). Although C:$N_{fr}$ has no direct influences on the
uptakes of water and nutrients, it can control the nitrogen required for fine root construction and the allocation of nitrogen to
roots and shoots, and hence the corresponding respirations (Raj *et al.*, 2014). These may explain why C:$N_{leaf}$ and C:$N_{fr}$
showed relatively high sensitivity for the DBT and the ENT in the temperate region, respectively. The analysis of
eco-physiological parameters also suggests that the parameters of SLA, FLNR, LFNR and $g_{CO_{2max}}$ may be consistently
influential, independent of sites or the type of sensitivity analysis. But the ranking of the parameters may vary according to
specific species and regions (Raj *et al.*, 2014).
In the sensitivity analysis, solar radiation, air temperature and humidity were identified as the essential meteorological
variables influencing GPP. Solar radiation determines the rate of photosynthesis which depends on the amount of absorbed
photosynthetically active radiation. Theoretically, increased solar radiation can promote photosynthesis by providing more





photosynthetic photon (Yu *et al.*, 2008). As radiation change is the most pronounced factor during cloudy conditions, in
which leaves are not saturated with light, unlike in full-sunny conditions (Yu *et al.*, 2008), our analysis showed that a
positive change in solar radiation during cloudy conditions could significantly promote vegetation growth. The simulations
of the BEPS and CLM-CN models showed that increased solar radiation resulted in higher GPP (Raczka *et al.*, 2013), which
was consistent with the simulations of our modified CNMM-DNDC model. Air temperature was identified as the secondly
sensitive meteorological variable as it influence the enzymatic activity of photosynthesis for mixed forests in both CBM and
DHM with broadleaved trees (Farquhar *et al.*, 1980). Previous studies have showed that ER is closely tied to temperature and
plant productivity (Yu *et al.*, 2008; Jagermeyr *et al.*, 2014). Similar effect of temperature on ER was also detected in the
sensitivity analysis of this study. Air temperature and solar radiation can substantially affect the ER by regulating enzymatic
activity and plant growth. Apart from the effects of solar radiation and air temperature, those of humidity were remarkable
for the ENT, as increased humidity caused declined vapour pressure deficit and promoted leaf-scale conductance, thus
resulting in higher GPP and lower ET (Sato *et al.*, 2015). In addition, the limited effects of soil properties on GPP and ER
suggest that the variations of GPP and ER among the three field sites could be primarily attributed to the latitudinal gradients
of meteorological variables, e.g., air temperature.

The sensitively analysis can quantify the contribution of input parameters/variables to the changes in model outputs,

which is widely applied in model calibration, diagnostic evaluation and uncertainty assessment (Pianosi *et al.*, 2016). In this
study, the one-at-a-time approach of sensitivity analysis was used to evaluate the influences of newly introduced
eco-physiological parameters and model inputs on the GPP and ER fluxes simulated by the modified CNMM-DNDC.
However, the one-at-a-time approach has two essential limitations. One is that only the effects of a specific parameter/input
variable can be provided, but those of the other parameters/inputs cannot be examined during each simulation. Another is
that the analysis may be invalid when investigated output variable is non-linearly tied with an investigated parameter/input
variable (Saltelli *et al.*, 2008). A global approach of sensitivity analysis can reflect the interactive effects of changes in
multiple parameters/inputs (Saltelli et al., 2000; Odongo *et al.*, 2013; Raj *et al.*, 2014). Thus, in order to identify the most
influential input parameters and provide insight into the model function, global sensitivity analysis is needed in future study
to comprehensively analyse the effects of multiple inputs, such as the variance-based sensitivity analysis.




**5 Conclusions**

The Catchment Nutrient Management Model - DeNitrification-DeComposition (CNMM-DNDC) is a lately developed process-oriented hydro-biogeochemical model. Its development is aiming at comprehensive simulation/prediction of multiple ecosystem variables (e.g., emissions/uptakes of carbon and/or nitrogen gases from terrestrial ecosystems, evapotranspiration, productivity, soil erosion, and slope runoff and catchment discharges of particle and/or dissolved carbon, nitrogen and phosphorous at plot, ecosystem, landscape, catchment/river basin, regional or global scales), which are often concerned in implementing the United Nations Sustainable Development Goals (SDGs) by 2030. However, the current version of this model uses the same simulation mechanism of plant growth for forest, grassland, wetland and cropland vegetation, thus resulting in large biases in simulating dynamical carbon fluxes of forests. In fact, the growth and turn over processes of forests are often different from other vegetation types. In this study, by referring to the Biome-BGC model, the current CNMM-DNDC model was updated through developing a special module on forest growth so as to improve its simulation mechanisms of forest vegetation. The modified processes of the updated model include photosynthesis, allocation, respiration, mortality and litter decomposition, which can detail the transformation and transportation of carbon and nitrogen in the plant-soil-water continuum of forests. Using the daily data observed by the eddy covariance systems at three typical forest sites located in different climate regions of the eastern Asia, the updated CNMM-DNDC model were calibrated and validated for simulating the gross primary productivity (GPP), ecosystem respiration (ER), net ecosystem carbon dioxide exchange (NEE) and evapotranspiration (ET) during the 8-years period of 2003–2010. The model can effectively reproduce the daily dynamics of the carbon and water fluxes, thus showing a strong capacity in capturing their annual variations. Compared with the original CNMM-DNDC, the updated model version generally performed much better in simulating the daily GPP, ER and ET fluxes, but had no remarkable improvement for the daily NEE. This comparison suggests the necessity of calibrating and validating each component of carbon fluxes to avoid the offsetting of model errors. The GPP and ER fluxes simulated by the updated model version are remarkable sensitive to the changes in canopy average specific leaf area, fraction of leaf nitrogen in Rubisco, annual leaf and fine root turnover fraction, maximum stomatal conductance and ratios of carbon to nitrogen in leaves and fine roots as the eco-physiological parameters and solar radiation, humidity and air temperature as the essential meteorological variables that regulate forest growth. However, the effects of soil properties on



the simulated GPP and ER were neglectable. The updated model was proven in this study to be capable of estimating the
carbon fluxes of various forest ecosystems. It thus provides a potential tool for quantifying multiple ecosystem variables in
close association with the SGDs, especially with improved credibility in simulating the carbon fluxes and budgets of forests.
However, the robust performances of the modified CNMM-DNDC in comprehensively simulating the aforementioned
ecosystem variables for addressing multiple SDGs at different scales still need verification and confirmation in further
studies.
**Code and data availability**
The source code and executive program of the improved model, as well as the input data can be obtained from
https://doi.org/10.5281/zenodo.13363688. All the observed data sets used in this study are available at the ChinaFLUX
(http://nesdc.org.cn/).
**Author contribution**
Zhang, W., Li, S., Li, Y. and Zheng, X. contributed to the idea and science of this study. Zhang, W. programmed the
newly developed module on forest growth, implemented the model simulations and data analysis and prepared the manuscript
with contributions from all co-authors. Han, S. collected and established the input database for modelling. Yu, G., Chen, Z.,
Wu, J., Wang, H. and Yan, J. contributed to observation and quality assurance/quality control of the measured data involved
in the updated model calibration and validation. Li, C., Yao, Z., Wang, R., Wang, K. and Chen, X. contributed to the
manuscript improvement.
**Competing interests**
The authors declare that they have no conflict of interest.
**Acknowledgement**
This study was jointly supported by the National Natural Science Foundation of China (42330607), the Chinese Academy
of Sciences (XDA23070100, ZDBS-LY-DQC007) and the National Key R&D Program of China (2022YFF0801904), and,





the National Key Scientific and Technological Infrastructure project "Earth System Science Numerical Simulator Facility"
(EarthLab).

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





Table 1 Statistics reflecting performances of original and updated CNMM-DNDC in simulating daily carbon fluxes of gross
primary productivity (GPP) and ecosystem respiration (ER) three forest sites.

| Item | Site | $n^a$ | NRMSE | | NSE | | Slope | | $R^{2d}$ | |
|---|---|---|---|---|---|---|---|---|---|---|
| | | | $Ori^b$ | $Mod^c$ | Ori | Mod | Ori | Mod | Ori | Mod |
| Daily GPP | $CBM_{cal}$ | 1826 | 0.94 | **0.35** | 0.19 | **0.89** | 0.77 | **0.94** | 0.29 | **0.90** |
| | $CBM_{val}$ | 1096 | 1.21 | **0.41** | -0.25 | **0.85** | 0.60 | **0.93** | 0.20 | **0.86** |
| | $QYZ_{cal}$ | 1826 | 0.51 | **0.35** | 0.02 | **0.55** | 0.82 | **0.85** | 0.20 | **0.68** |
| | $QYZ_{val}$ | 1096 | 0.47 | **0.33** | 0.18 | **0.59** | 0.55 | **0.88** | 0.65 | **0.67** |
| | $DHM_{cal}$ | 1826 | 0.64 | **0.35** | **-1.57** | **0.16** | 0.75 | **0.92** | - | **0.21** |
| | $DHM_{val}$ | 1096 | 0.64 | **0.38** | **-1.68** | **0.08** | 0.73 | **0.89** | - | **0.18** |
| | $All_{cal}$ | 5478 | 0.71 | **0.35** | 0.03 | **0.76** | 0.79 | **0.90** | 0.19 | **0.79** |
| | $All_{val}$ | 3288 | 0.80 | **0.37** | -0.22 | **0.74** | 0.71 | **0.90** | 0.11 | **0.79** |
| Daily ER | $CBM_{cal}$ | 1826 | 0.69 | **0.21** | 0.25 | **0.93** | 0.83 | **1.00** | 0.33 | **0.93** |
| | $CBM_{val}$ | 1096 | 0.94 | **0.27** | -0.35 | **0.89** | 0.65 | **1.08** | 0.11 | **0.90** |
| | $QYZ_{cal}$ | 1826 | 0.57 | **0.26** | -0.53 | **0.68** | 0.83 | **0.93** | - | **0.71** |
| | $QYZ_{val}$ | 1096 | 0.52 | **0.21** | -0.41 | **0.77** | 0.45 | **1.05** | 0.40 | **0.78** |
| | $DHM_{cal}$ | 1826 | 0.59 | **0.24** | -1.32 | **0.62** | 0.73 | **0.92** | - | **0.68** |
| | $DHM_{val}$ | 1096 | 0.51 | **0.18** | -0.76 | **0.78** | 0.83 | **1.00** | - | **0.78** |
| | $All_{cal}$ | 5478 | 0.64 | **0.24** | 0.03 | **0.87** | 0.81 | **0.96** | 0.17 | **0.87** |
| | $All_{val}$ | 3288 | 0.69 | **0.23** | -0.34 | **0.85** | 0.77 | **1.05** | - | **0.86** |
| Daily NEE | $CBM_{cal}$ | 1826 | 3.79 | **2.85** | 0.02 | **0.45** | 0.60 | **0.71** | 0.15 | **0.55** |
| | $CBM_{val}$ | 1096 | 3.55 | **2.47** | -0.09 | **0.47** | 0.50 | **0.69** | 0.17 | **0.61** |
| | $QYZ_{cal}$ | 1826 | **0.95** | 1.17 | **0.35** | 0.00 | 0.83 | **0.64** | **0.40** | 0.32 |
| | $QYZ_{val}$ | 1096 | **1.28** | 1.59 | **0.29** | -0.09 | 0.75 | **0.57** | **0.41** | 0.34 |
| | $DHM_{cal}$ | 1826 | 1.13 | **1.05** | **-0.20** | -0.04 | 0.80 | **0.87** | - | **-** |
| | $DHM_{val}$ | 1096 | 1.83 | **1.71** | **-0.28** | -0.11 | 0.55 | **0.63** | - | **0.02** |
| | $All_{cal}$ | 5478 | 1.49 | **1.39** | 0.13 | **0.24** | 0.75 | **0.70** | 0.19 | **0.38** |
| | $All_{val}$ | 3288 | 2.04 | **1.84** | -0.01 | **0.18** | 0.60 | **0.63** | 0.19 | **0.40** |
| Daily ET | $CBM_{cal}$ | 1826 | 0.48 | **0.46** | 0.74 | **0.76** | 1.09 | **1.07** | 0.76 | **0.77** |
| | $CBM_{val}$ | 1096 | 0.47 | **0.42** | 0.76 | **0.81** | 0.97 | **0.99** | 0.76 | **0.81** |
| | $QYZ_{cal}$ | 1826 | 0.83 | **0.40** | -0.31 | **0.70** | 0.69 | **0.98** | 0.08 | **0.70** |
| | $QYZ_{val}$ | 1096 | 0.70 | **0.32** | 0.00 | **0.80** | 0.50 | **1.07** | 0.41 | **0.81** |
| | $DHM_{cal}$ | 1826 | 0.72 | **0.42** | -0.57 | **0.47** | 0.73 | **0.96** | - | **0.47** |
| | $DHM_{val}$ | 1096 | 0.68 | **0.35** | -0.55 | **0.59** | 0.72 | **0.93** | - | **0.62** |
| | $All_{cal}$ | 5478 | 0.74 | **0.43** | 0.03 | **0.68** | 0.76 | **0.99** | 0.22 | **0.68** |
| | $All_{val}$ | 3288 | 0.68 | **0.35** | 0.18 | **0.78** | 0.96 | **1.00** | 0.48 | **0.78** |

CBM, QYZ and DHM represent the forest site names of Changbai Mountains, Qianyanzhou and Dinghu Mountains,
respectively. The subscripts of cal and val indicate the results during the periods of calibration (in 2003–2007) and validation
(in 2008–2010), respectively, for all the sites. *n* denotes the number of daily or annual observations. Ori and Mod columns
contain the statistics on simulations of the original and updated CNMM-DNDC, respectively. NRMSE, NSE, Slope and $R^2$
column contain statistics of normalized root mean square error, Nash-Sutcliffe efficiency, and the slope and determination
coefficient of zero-intercept linear regression.





Table 2 Statistics reflecting performances of original and updated CNMM-DNDC in simulating annually cumulated carbon
fluxes of gross primary productivity (GPP) and ecosystem respiration (ER) across all three forest sites.

| Item | Site | $n$ | NRMSE | | NSE | | Slope | | $R^2$ | |
|---|---|---|---|---|---|---|---|---|---|---|
| | | | Ori | Mod | Ori | Mod | Ori | Mod | Ori | Mod |
| Annual GPP | All$_{cal}$ | 15 | 0.10 | **0.07** | 0.04 | **0.61** | 1.08 | **0.98** | 0.51 | **0.64** |
| | All$_{val}$ | 9 | 0.13 | **0.06** | 0.04 | **0.79** | 0.99 | **0.98** | 0.04 | **0.80** |
| Annual ER | All$_{cal}$ | 15 | 0.13 | **0.10** | 0.40 | **0.67** | 1.08 | **0.98** | 0.60 | **0.68** |
| | All$_{val}$ | 9 | 0.19 | **0.12** | -0.69 | **0.33** | 1.07 | **1.08** | - | **0.62** |
| Annual NEE | All$_{cal}$ | 15 | 0.23 | **0.26** | 0.66 | **0.55** | 1.06 | **0.98** | 0.69 | **0.55** |
| | All$_{val}$ | 9 | 0.48 | **0.51** | -0.71 | **-0.96** | 0.74 | **0.71** | 0.19 | **0.28** |
| Annual ET | All$_{cal}$ | 15 | 0.17 | **0.13** | 0.29 | **0.61** | 0.96 | **0.99** | 0.34 | **0.61** |
| | All$_{val}$ | 9 | 0.07 | **0.08** | 0.89 | **0.88** | 0.99 | **0.98** | 0.89 | **0.89** |

Meanings of all abbreviations and subscripts, as well as year periods for the calibrations and validations are found in the
footnotes of Table 1.





**Figures**

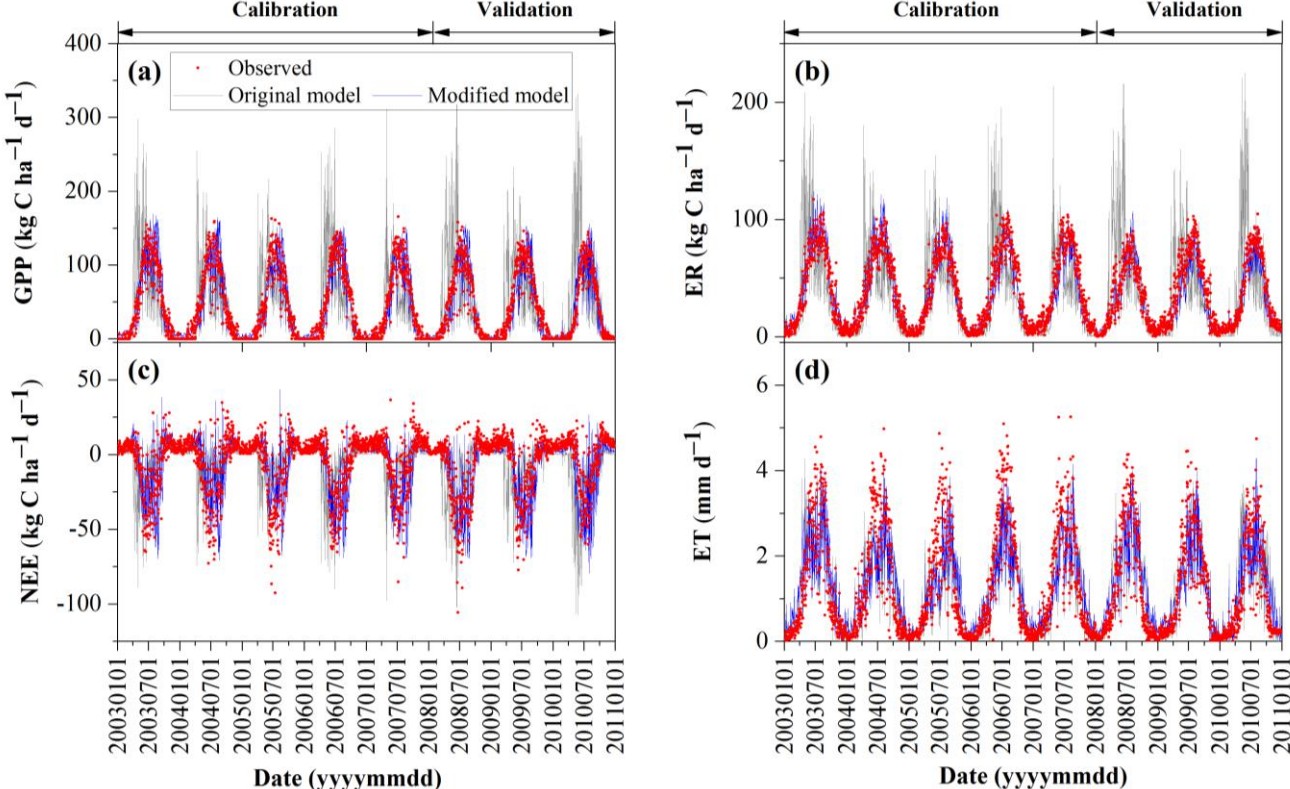

Figure: 1 Observations and simulations of original and modified CNMM-DNDC model on daily carbon fluxes of gross primary productivity (GPP), ecosystem respiration (ER) and net ecosystem-atmosphere exchange of carbon dioxide (NEE) and on daily water vapour fluxes of evapotranspiration (ET) from temperate mixed forest of evergreen needle leaf and deciduous broad leaf trees at the Changbai Mountain (CBM) site. The legends in panel a apply for other panels.



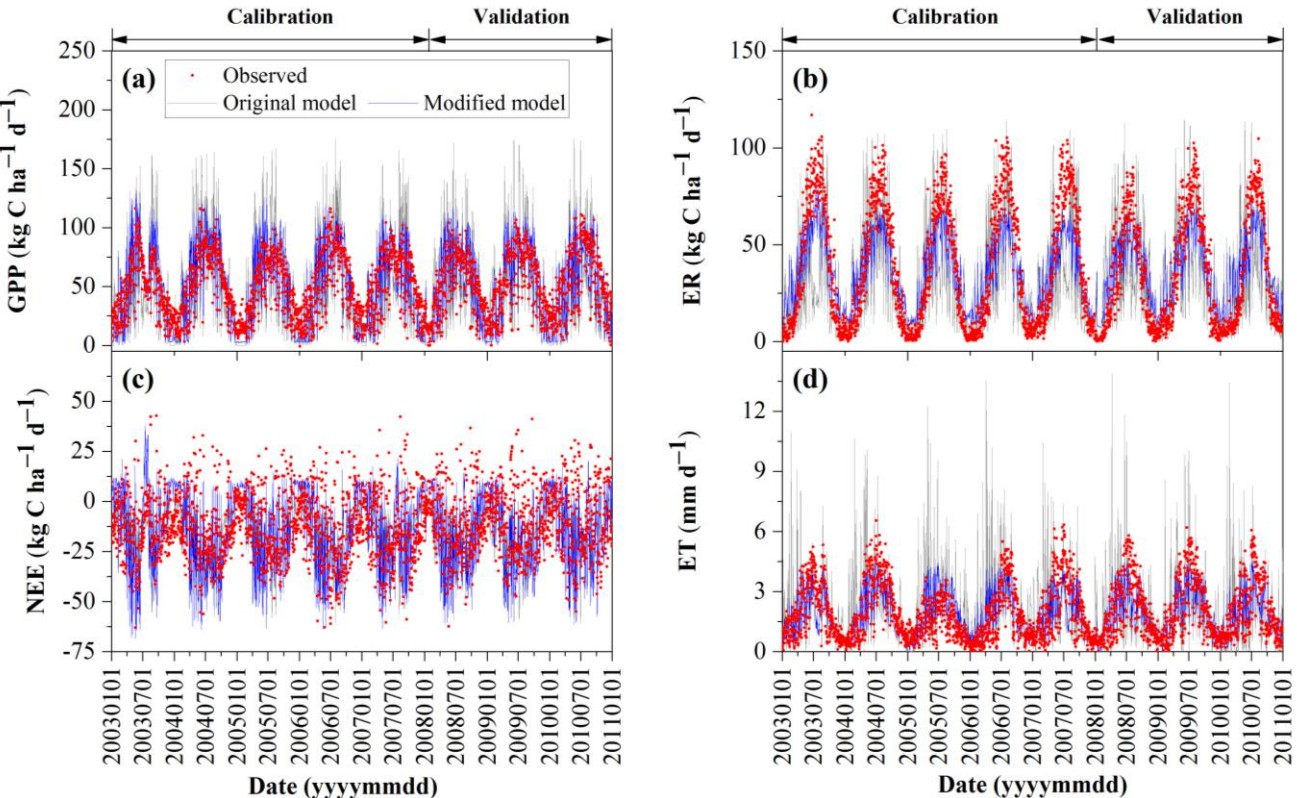

**Figure 2: Observations and simulations of original and modified CNMM-DNDC model on daily carbon fluxes of gross primary productivity (GPP), ecosystem respiration (ER) and net ecosystem-atmosphere exchange of carbon dioxide (NEE) and on daily water vapour fluxes of evapotranspiration (ET) from subtropical evergreen needle leaf forest at the Qianyanzhou (QYZ) site. The legends in panel a apply for all other panels.**

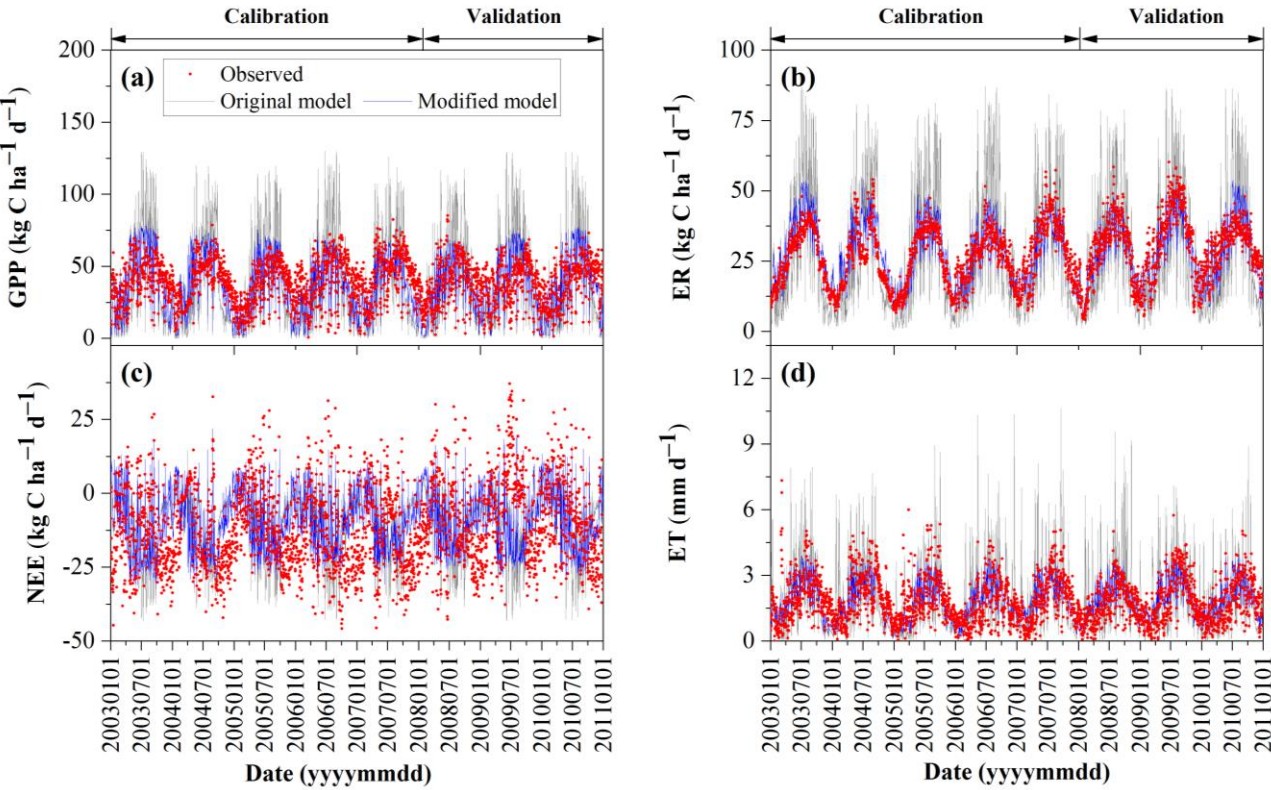

Figure 3: Observations and simulations of original and modified CNMM-DNDC model on daily carbon fluxes of gross primary productivity (GPP), ecosystem respiration (ER) and net ecosystem–atmosphere exchange of carbon dioxide (NEE) and on daily water vapour fluxes of evapotranspiration (ET) from subtropical mixing forest of evergreen needle and evergreen broad leaf trees at the Dinghu Mountains (DHM) site. The legends in panel a apply for all other panels.





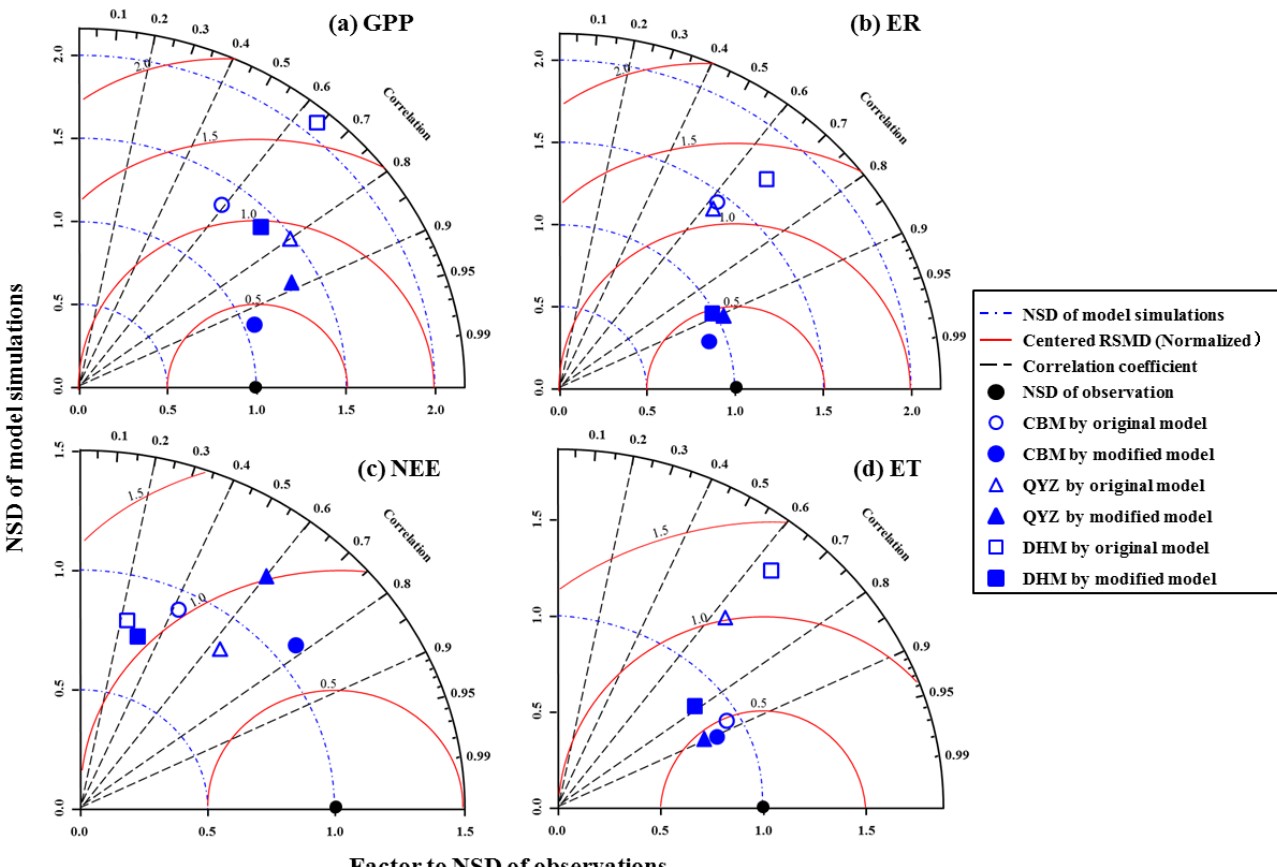

**Figure: 4 Performance of original (empty grey symbols) and modified (solid blue symbols) model versions compared to observations (black solid points) at each forest site in validation years (2008–2010). Statistics shown in the Taylor diagrams are for daily simulated and observed carbon fluxes of gross primary productivity (GPP), ecosystem respiration (ER) and net ecosystem-atmosphere exchange of carbon dioxide (NEE), and water vapour fluxes of evapotranspiration (ET). The full site names of CBM, QYZ, and DHM are referred to the footnotes of Table 1.**





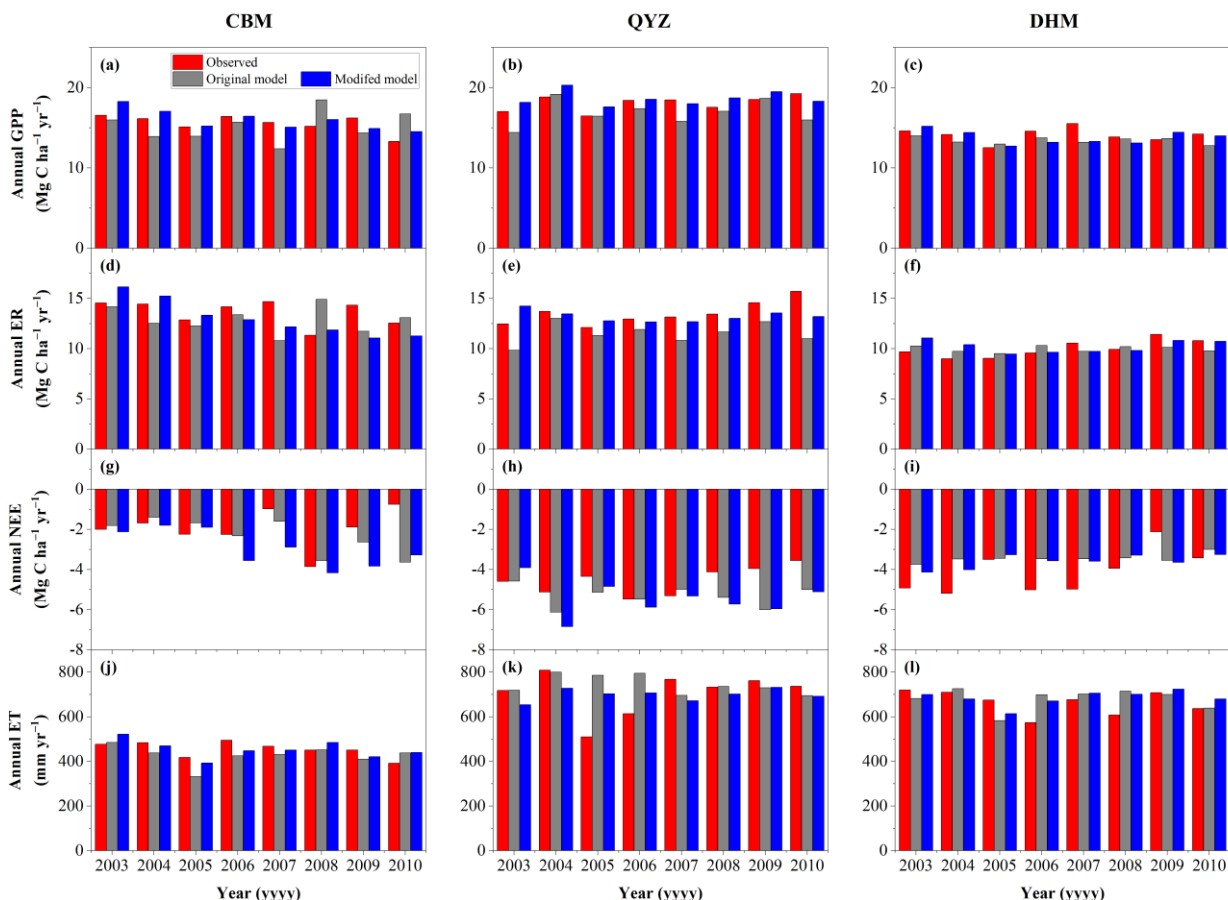

944

**Figure: 5 Observed and simulated annual carbon fluxes of gross primary productivity (GPP), ecosystem respiration (ER) and net ecosystem-atmosphere exchange of carbon dioxide (NEE) and water vapour fluxes of evapotranspiration (ET) at three forest sites. The full site names of CBM, QYZ and DHM are referred to the footnotes of Table 1. The legends in panel a apply for all other panels. The simulations were provided by the original and modified CNMM-DNDC model.**



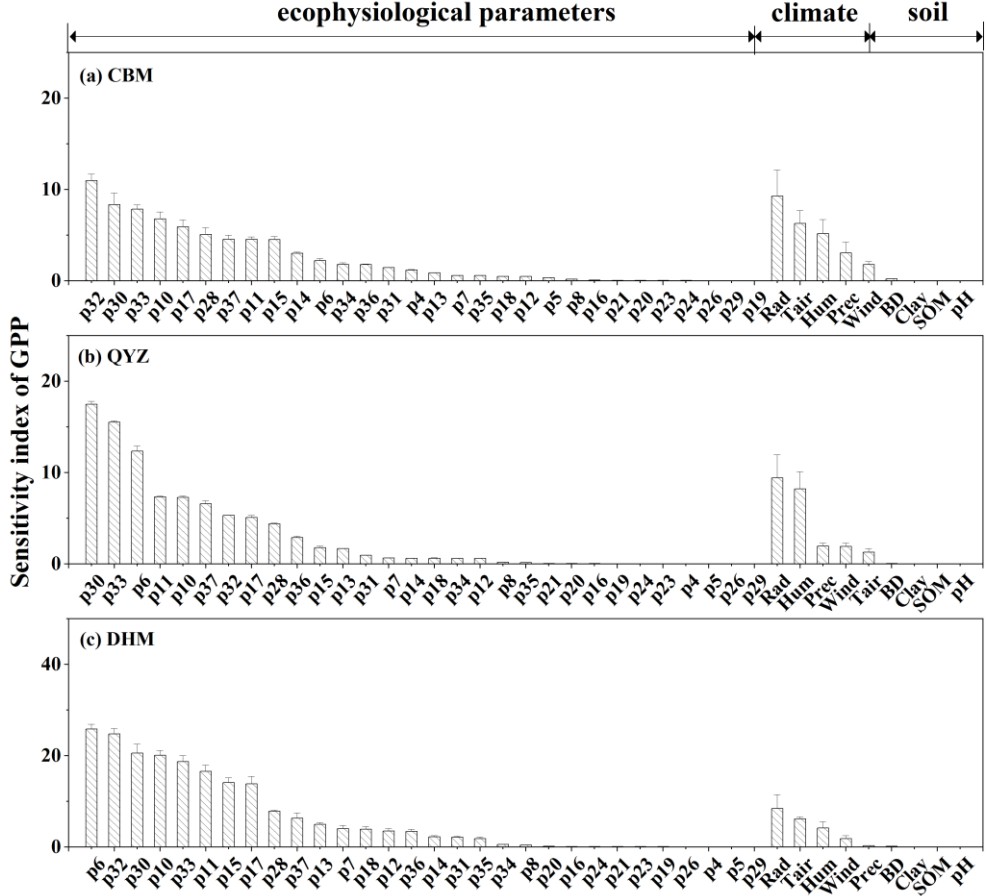

**Figure: 6** Sensitivity indexes of modified CNMM-DNDC simulations on gross primary productivity responding to alternations in individual eco-physiological parameters and model inputs of meteorological variables and soil properties at each site. An index is given as the mean (a wide bar) and standard deviation (an error bar) of three indexes, each resulted from the simulations for one year. The parameter name and value for each parameter are referred to Table S2.



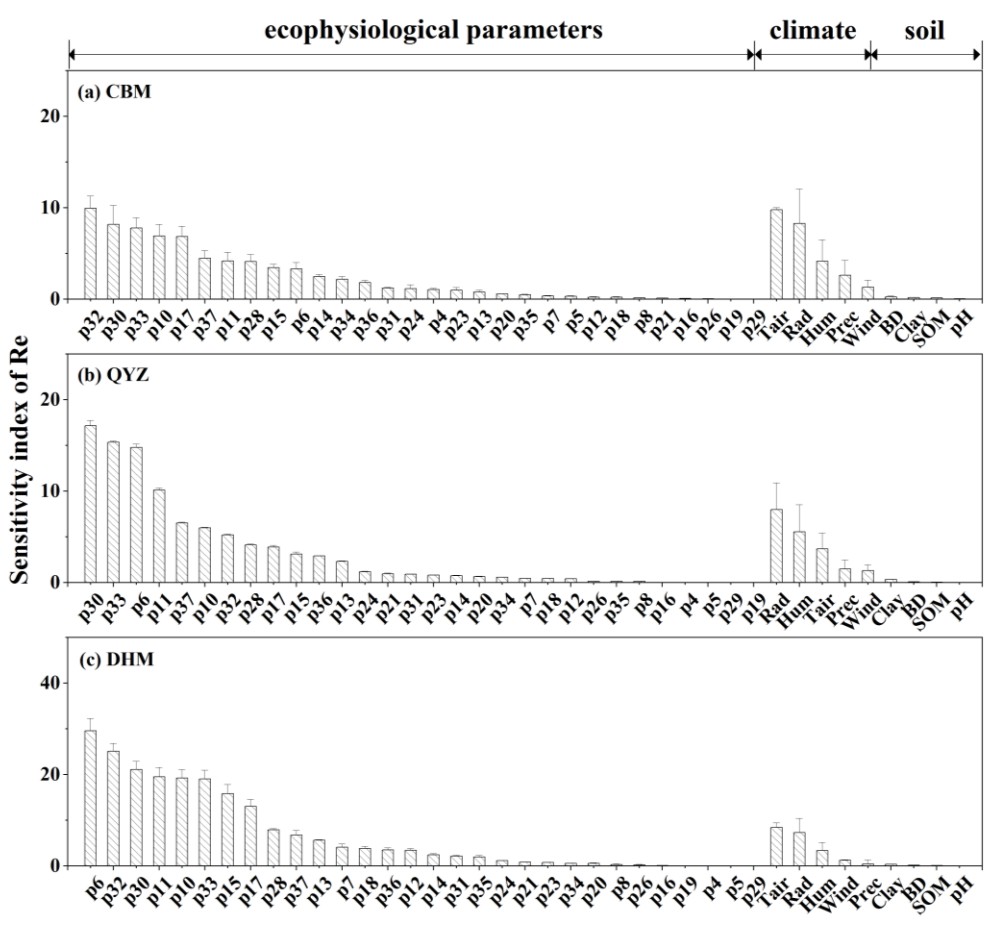

954

**Figure: 7 Sensitivity indexes of modified CNMM-DNDC simulations on ecosystem respiration responding to alternations in individual eco-physiological parameters and model inputs of meteorological variables and soil properties at each site. An index is given as the mean (a wide bar) and standard deviation (an error bar) of three indexes, each resulted from the simulations for one year. The parameter name and value for each parameter are referred to Table S2.**