# Peer review of "An improved hydro-biogeochemical model (CNMM-DNDC V6.0) for"

_Geoscientific Model Development, 2024_

## Author Comment (AC1)

In this paper, Zhang et al. described and evaluated a new version of the hydro-biogeochemical model CNMM-DNDC that includes a more detailed representation of processes driving carbon and water fluxes between the vegetation and the atmosphere at forest sites. This includes $CO_2$ and water exchanges at the leaf level, carbon allocation and plant growth and plant mortality. I believe this could be a valuable contribution to the corresponding modelling community. I have several important comments or suggestions however.

First, some important processes or modelling choices are not described or explained. For example how the stomatal and boundary layer conductances to water vapor are computed is not provided. However this is quite key for both carbon and water fluxes. Why mortality rates is constant (if I have well understood) and provided as input is not explained/discussed, while part of mortality could /should be environmentally driven, especially to make predictions under varying conditions.

**Revised**.

The calculation of the conductance to the water vapour has been detailed in the Text S1. "*The total leaf conductance to water vapor is calculated by combining the stomatal ($g_s$), boundary ($g_{bl}$), and cuticle ($g_c$) level conductance in parallel for sun and shade leaves, respectively (Eq. 1−4). The maximum rate of stomatal ($g_{smax}$), boundary ($g_{blmax}$), and cuticle ($g_{cmax}$) level conductance are user defined eco-physiological parameters for different forest types detailed in Table S3. A conductance correction factor (gcorr, dimemsionless) is calculated for the current air temperature ($T_{air}$, °C) and atmospheric pressure (pa, Pa) (Eq.5).The cuticle and boundary layer conductance are only scaled by gcorr, but the stomatal conductance is also scaled by a series of multipliers between 0 and 1 (Eq. 6; f), which are photosynthetic photon flux density ($f_{PPFD}$), soil water potential ($f_{SM}$), minimum temperature ($f_{Tair}$) and vapor pressure deficit ($f_{VPD}$). The $f_{PPFD}$ is a function of photosynthetic photon flux density (PPFD, µmol $m^{-2}$ $s^{-1}$), projected leaf area index of whole canopy (PLAI, $m^2$ $m^{-2}$) and PPFD for 50% stomatal closure (75 µmol $m^{-2}$ $s^{-1}$), which is calculated for sun and shade leaves, respectively (Eq. 7). The others are calculated based on the current, maximum and minimum values of the variables for all leaves.*" (**See Text S1 in the revised supplementary materials**).

The discussion about the mortality has been added as the reviewer suggested. "*In the updated model, user-defined mortality rates were used in the simulation. Recent studies showed that the mortality rates at stand level are regulated by various factors, such as the stand age and density, tree basal area, stand squared mean diameter at breast height, standardized precipitation evapotranspiration index, drought length, mean annual temperature and precipitation, elevation and slope (Subedi et al., 2021; Yan et al., 2024). The count-data models combined with random effects of the survey plots have been developed to simulate the tree mortality at different sites (Zhang et al., 2015; Yan et al., 2024). However, the general parameterizations of mortality rates, including the factors of stand, climate and topography, are still not available for the*

*process-oriented models due to knowledge gaps. Thus, in order to predict the carbon cycles of forest using the process-oriented models, more attention should be focused on the simulation of tree mortality under varied environmental conditions and disturbances.*" (**See lines 564-573 in the revised manuscript**).

Second, using a range of metrics, the study thoroughly highlights the improved performance of the new modified model over the original one in simulated carbon and water fluxes at daily scale. However, it is hard to correctly interpret whether this improved performance truly serves the model predictive objective as nothing is said about the calibration. How many parameters were calibrated in the new version of the model ? how many parameters were calibrated in the original version of the model? Does the improved performance result from an increased number of calibrated parameters? How were the calibration performed actually? Which parameters were calibrated against which variables/data? How transferable is the model to other sites and/or environmental conditions? This has to be described and discussed.
**Revised**.
The statements about model calibration have been added. "*The required parameters for forest simulation, including forest type, carbon contents of leaf and stem and some of eco-physiological parameters, were primarily adapted from the field observations provided by the NESDC or from the peer reviewed literatures (Li, 2018; Li, 2019; Fang, 2022). The other eco-physiological parameters referred to the default values (Table 1). The parameter of fraction of leaf nitrogen in Rubisco (p32) was calibrated using the normalized root mean square error (NRMSE) between observed and simulated carbon and water fluxes during 2003–2007. The upper and lower boundaries of the parameter value (p32) were set as twice and half of the default value. The parameter was identified when the value of NRMSE was the minimum.*" (**See lines 326-332 in the revised manuscript**).The sources of the eco-physiological parameters have been marked in the revised Table 1, indicating the values from literatures or calibration, as well as the range of parameters used for calibration. (**See Table 1 in the revised manuscript**).
The discussions about the reasonable prediction of the updated model have been added as reviewer suggested. "*According to Table 1, except for the CBM site with sufficient localized parameters, the eco-physical parameters in the other two sites primarily came from default values. The comparable model performances at the three sites indicated the applicability of the updated model without various observations and comprehensive calibration. But the localization of eco-physical parameters can improve the model performances without doubt.*" (**See lines 561-564 in the revised manuscript**).

Also, the model calibration and evaluation relied exclusively on eddy flux data. More details are required on this data. How was the partitioning between GPP and ER performed? What are the uncertainties in those estimates? As model performance is better for GPP and ER with the modified than original model, but not for NEE, that would be an interesting point to discuss.

**Revised**.

The details about eddy flux data have been added in the Text S2. "*As the three sites are all members of Chinese Terrestrial Ecosystem Flux Observation and Research Network (ChinaFLUX), the quality control and processing of flux data were carried out based on standardized approaches (Yu et al., 2008), including three-dimensional rotation of the coordinate, correction for the variation of air quality, removal of abnormal values, check of energy balance closure, etc. The data gaps were filled mainly by means of the nonlinear regression method. For small gaps (< 2h), the missing data were linearly interpolated. Larger gaps, such as daytime and night-time gaps, were treated separately when filling the gaps in the $CO_2$ data sets. The missing daytime flux data were estimated as a function of photosynthetic photon flux density using the Michaelis-Menten equation with a 10-day moving window. To estimate the gross primary productivity (GPP), the ecosystem respiration (ER) of day was estimated with the relationships between the ER of night versus soil temperature and water content (Yu et al., 2008). The ER was sum of corresponding values of day and night. GPP was estimated as the sum of ER and $CO_2$ flux.*" (**See Text S2 in the revised supplementary materials**).

The discussion about simulated NEE has been added. "*At the annual scale, the correlation between simulated and observed GPP was consistently higher than that of NEE, indicating the high sensitivity of NEE to small relative errors in large GPP fluxes (Raczka et al., 2013). Meanwhile, the correlation of ER between simulations and observations was lower than that of GPP, suggesting that ER may be the main contributor to the poor simulation of inter-annual variability in NEE. Although process-oriented models can effectively simulate the different types of carbon fluxes by incorporating the inter-annual influences of temperature and soil moisture, the simulated NEE in this study, as well as others (Keenan et al., 2012; Raczka et al., 2013), can draw a conclusion that process-oriented models do not adequately explain the observed inter-annual variability in NEE, yet.*" (**See lines 545-552 in the revised manuscript**)

Finally, although not a native English speaker myself, I strongly recommend checking the English and phrasing throughout the manuscript, particularly in the introduction.
**Revised**.
The English usage has been revised throughout the manuscript.

Point-by-point comments :

Title : what do you mean exactly by "typical" (same l. 32)? could/should "evapotranspiration" be replaced by "water" to follow the same structure as for "carbon"?
**Revised**.
The title has been revised as the reviewer suggested (**See line 3 in the revised manuscript**). The "typical" indicates the widely distribution in the corresponding

regions and the statements have been revised throughout the manuscript. "......*three typical forest sites which are widely distributed in the subtropical and temperate climate regions in eastern Asia (2003–2010)......*" (**See lines 32-33 in the revised manuscript**).

33, 97, 100, 102, 281 etc…: prefer "evaluation"/"evaluated" instead of "validation"/"validated"
**Revised**.
The statements have been revised throughout the manuscript.

35: this is a bit hard to follow and suggest rephrasing. Also does a negative value here (e.g. -6%) means that the NRMSE actually increased between the previous version and the version presented here ? It is a bit unclear how the NRMSE of ET can be reduced by 38% at daily scale, but increased by 3% at annual scale…
**Revised**.
The sentences have been revised to avoid confusion (**See lines 35-36 in the revised manuscript**). The explanations have been added in the section of 3.1. "*Although the simulated ET at the daily scale by the original model showed significant deviations with much more intensive variations, especially for sites of QYZ and DHM, the trade-off effects of extreme values for the original models led to comparable performances of both models in simulating annual ET (Fig. 2j−l).*" (**See lines 458-461 in the revised manuscript**)

55: "in comparison" to what ? I am not sure I would oppose data and models, both are needed and complementary (and without data, numerical models would be far less advanced).
**Revised**.
The sentence has been revised as the reviewer suggested. "*The numerical models are promising tools to combine data from different sources and characterize the vegetation and soil processes more completely.*" (**See lines 55-56 in the revised manuscript**).

59: I am not sure this defines "process-based models" and would rephrase this sentence. At least specify "process-based models "of what.
**Revised**.
The sentence has been revised as the reviewer suggested. "*...... and process-oriented models, the last type of model is an important scientific tool that was established based on basic theories of physics, chemistry, and biogeochemistry processes*" (**See lines 58-59 in the revised manuscript**).

89: it would be great to provide the reader with at list a brief description of the specificities of the CNMM-DNDC model: in what respect is it different (or not) with the other hydro-biogeochemical models, especially the ones mentioned in the previous paragraph?

**Revised**.

The description of the key specificity of the CNMM-DNDC model has been added based on the comments. "*Compared with the models above mentioned, the key specificity of CNMM-DNDC is the realization of modeling horizontal transportations of water and nutrients in both soil surface and profile from grid to grid, which supports the simulation of hydro-biogeochemical processes at the catchments.*" (**See lines 88-90 in the revised manuscript**).

96: Zhang et al. 2018 is repeated twice.
**Revised**.
The citations have been revised (**See lines 85-93 in the revised manuscript**).

97: "simultaneously" should be moved elsewhere.
**Revised**.
The sentence has been rephrased (**See line 94 in the revised manuscript**).

107: unclear why mortality is related to the simplified representation of biomass allocation here, can you specify? This is better distinguished l. 160-161.
**Revised**.
The sentences have been rewritten to make it clear. "*Such simplification may induce large uncertainties, as photosynthate allocation is substantially important for accurately simulating the carbon and nitrogen cycles of forest ecosystems.*" (**See lines 102-104 in the revised manuscript**).

115 (also 118): what do you mean by "typical" ? what is a "typical" forest? What do the three sites have in common, and what make them different?
**Revised**.
The "typical" means the widely distribution and representativeness of forest types in the corresponding regions. The sentences have been rewritten to make it clear and the descriptions have been revised throughout the manuscript. "*......in the typical forest ecosystems of the eastern Asia which are widely distributed in the corresponding regions.*" (**See line 302 in the revised manuscript**).

119: if I understood this correctly this should be phrased the other way around: simulated outputs (GPP, ER, etc…) are sensitive to some parameter values/model inputs.
**Revised**.
The related sentences have been revised throughout the manuscript as the reviewer suggested. "*(iii) identify the eco-physiological parameters and model inputs which can substantially influence the simulated GPP and ER of the examined forests using different methods.*" (**See lines 122-124 in the revised manuscript**).

130: what a "comprehensive function " is unclear to me.
**Revised**.

The sentences have been rephrased to make it clear. The previous "comprehensive function" indicated the extension of model's functions to improve the model's abilities. *"Its later versions were established through several updates to extend its functions and improve the universally applicability"* (**See lines 132-133 in the revised manuscript**).

137: "it has realized systematic simulation of ": what you mean here is unclear to me.
**Revised**.
The sentences have been revised. The "systematic simulation" indicates the simulation for the continuum of atmosphere, vegetation, soil, and water. *"The model regards the atmosphere, vegetation, soil, and water as a continuum and simulates the tightly coupled carbon, nitrogen and water cycles of the continuum at the catchment based on basic theories of physics, chemistry, and biogeochemistry."* (**See lines 134-136 in the revised manuscript**).

139: "be user-defined" or "be defined by the user"
**Revised**.
The sentences have been revised. *"The simulated soil depth and temporal and spatial resolutions of the model are all allowed to be defined by the user depending on the availability of driving data and/or research objectives."* (**See lines 142-143 in the revised manuscript**).

140: is 4m a hard constraint? What prevents from simulating deeper soil (which might be relevant in some systems)?
**Revised**.
The sentences have been revised to make it clear. "4m" is not a hard constraint, but module of groundwater circulation is not included in the CNMM-DNDC. *"The simulated soil profile could be down to 4 m deep or deeper, but the module of groundwater circulation is not included."* (**See lines 144-145 in the revised manuscript**).

161 (alos l. 107): it is not 100% clear what you mean by "photosynthetic allocation", do you mean "photosynthate allocation"?
**Revised**.
The "photosynthetic allocation" has been replaced by "photosynthate allocation" throughout the manuscript to make it clear.

168: "mortality of forests" or "mortality of trees"? What is the biological resolution of this new module? Is it individual-based, cohort-based, stand-based?
**Revised**.
The sentences have been revised to make it clear. This module can be thought as an estimate of stand level processes. *"This module can be thought as an estimate of stand level processes that have been aggregated and averaged to a per unit area basis and can simulate the processes of photosynthesis, litter decomposition, photosynthate*

*allocation, respiration and mortality of forests.*" (**See lines 171-174 in the revised manuscript**).

171: "live stems": does it refer to sapwood or to both sapwood and hartwood of living trees? Similarly, "dead stems": does it refer to heartwood or to both sapwood and heartwood of dead trees? (Same question for live/dead coarse roots). If the latter not sure why this is distinguished from a woody debris pool? Also distinction between sapwood and heartwood is needed to correctly represent stem respiration for example. **Revised**.

The sentence has been added to detail the contents. The live stems and coarse roots can be regarded as the sapwood, while the dead stems and coarse roots is the heartwood. The respiration was calculated for live and dead tissues, respectively. "*The live stems and coarse roots can be regarded as the sapwood, while the dead stems and coarse roots is the heartwood. The coarse woody debris pool is the first pool that dead coarse roots and dead stem wood enter when they die. This pool then fragments into the litter pools over time.*" (**See lines 177-180 in the revised manuscript**).

193: how is the conductance to (and not "of") water vapour gH20 computed ? does it include both the stomatal and leaf boundary layer conductance ? this is key. Humidity and wind speed are probably used for that computation ? Are those variables considered to be the same for sun and shade leaves? **Revised**.

The details about the calculation of the conductance to the water vapour ($g_{H_2O}$) has

been added in the supplementary materials. "*As the three sites are all members of Chinese Terrestrial Ecosystem Flux Observation and Research Network (ChinaFLUX), the quality control and processing of flux data were carried out based on standardized approaches (Yu et al., 2008), including three-dimensional rotation of the coordinate, correction for the variation of air quality, removal of abnormal values, check of energy balance closure, etc. The data gaps were filled mainly by means of the nonlinear regression method. For small gaps (< 2h), the missing data were linearly interpolated. Larger gaps, such as daytime and night-time gaps, were treated separately when filling the gaps in the $CO_2$ data sets. The missing daytime flux data were estimated as a function of photosynthetic photon flux density using the Michaelis-Menten equation with a 10-day moving window. To estimate the gross primary productivity (GPP), the ecosystem respiration (ER) of day was estimated with the relationships between the ER of night versus soil temperature and water content (Yu et al., 2008). The ER was sum of corresponding values of day and night. GPP was estimated as the sum of ER and $CO_2$ flux.*" (**See Text S1 in the revised supplementary materials**).

197: in some systems, the co-limitation of photosynthetic capacities by leaf phosphorous content might be important (Domingues et al. 2010; Walker et al. 2014). Is it the case in the study systems?

**Reponses**.

In the current version, the co-limitation of photosynthetic capacities by leaf phosphorous has not been included, which could be incorporated in future study. The discussion about this has been added. "*In this version, the co-limitation of photosynthetic capacities by leaf phosphorous has not been considered.*" (**See lines 208-209 in the revised manuscript**). "*Many studies also emphasized the co-limitation of photosynthetic capacity by leaf nitrogen and phosphorus which has been identified as globally important determinants (Domingues et al., 2010; Walker et al., 2014). Thus, the effects of leaf phosphorus should be considered in the development of the process-oriented models.*" (**See lines 521-523 in the revised manuscript**).

203, 208: where these values come from ?
**Revised**.
The reference has been added as the reviewer suggested (**See lines 24 and 219 in the revised manuscript**).

223: how is the proportion of nitrogen retranslocated determined?
**Revised**.
The sentence has been revised to make it clear. "*......while the nitrogen removed from the leaves before senescence is re-translocated based on the ratio of carbon to nitrogen of leaf litter.*" (**See lines 233-234 in the revised manuscript**).

229-230: the rationale supporting the fact that "the actual decomposition [is] scaled depending on the competing plant nitrogen demand during allocation" is unclear to me, can you elaborate ? and how is the plant nitrogen demand during allocation determined ?
**Responses**.
The detailed descriptions about the actual litter decomposition rate were presented in the revised version. "*The immobilized nitrogen by microbes in litter decomposition is provided by soil available mineral nitrogen pool which also offers the nitrogen required by plant growth. If the soil available mineral nitrogen cannot satisfy the demands of plant growth and potential litter decomposition, the actual decomposition rate would be scaled based on the fractions of two components (plant growth and litter decomposition) and soil available mineral nitrogen, which has been detailed during allocation in the section of 2.1.2.4.*" (**See lines 247-251 in the revised manuscript**).

236-236: if soil water pressure refers to soil water potential, I would suggest to replace "pressuer" by "potential" and replace Minpressure by , Satpressure by , and same for SMpressure, which are much more common notation.
**Revised**.
The descriptions of equations have been revised as reviewer suggested. "*......with the calculated soil water potential under saturation. The minimum soil water potential*

*(Minpotential) was set as −10 Mpa.*" (**See line 246 in the revised manuscript**).

236: In this section, I found it particularly hard to identify what is a variable updated/computed by the model at each timestep from what is a fixed parameter (user-defined or not) or a constant. An additional column in table S2 providing the symbol used in the main text would help. And a similar table with variable could be useful.
**Revised**.
The Table S4 about litter decomposition has been updated to make it clear. "*The fractions related to the decomposition processes of leaf, fine root, stem and litters, as well as the maximum rate constants and biomass loss through heterotrophic respiration, are all defined as constants (Table S4).*" (**See Table S4 and lines 243-244 in the revised supplementary materials**).

244: where does the carbon available for allocation come from of there is a carbon pool deficit? Are there carbon reserves? This is not mentioned l. 171.
**Revised**.
The statement has been revised to make it clear. "*If the difference is negative, it means a carbon pool deficit. The repayment of this carbon pool deficit is calculated in the following time steps so that the deficit is over in one year and the available carbon for allocation is first allocated to alleviate the deficit.*" (**See lines 257-259 in the revised manuscript**).

245: can you explain why?
**Revised**.
The explanation has been added as reviewer suggested. "*All new allocations to other organs or tissues are constrained by the new leaf carbon allocation due to the allocation priorities (Waring and Running, 2007). As under stressed conditions, trees tend to modify the allocation of carbohydrates so that new leaf is favoured over the production of stem growth (Waring and Pitman, 1985).*" (**See lines 259-262 in the revised manuscript**).

266: what does the 8 (denominator) represent in equ. 28? Unit? Same for equ. 29.
**Revised**.
The details have been added to make it clear. "*......with the conversion coefficient (8.0) from daily level to 3 hours*" (**See line 283 in the revised manuscript**). "*......with the conversion coefficient (86400) from daily level to seconds*" (**See line 288 in the revised manuscript**).

267: "relationship" à "slope". Is this value common to all tissues?
**Revised**.
The statement has been revised to make it clear. "*......with a relationship of 0.218 kg C d$^{-1}$ kg$^{-1}$ N for all the calculated tissues (Ryan, 1991)*" (**See lines 284-285 in the revised manuscript**).

277: if mortality rates are fully user-defined (and fixed throughout simulations), I wonder how the model could establish predictions under varying environmental conditions and disturbances (e.g. climate change is mentioned as a motivation in the introduction). Although I fully acknowledge the difficulties to simulate mortality through process-based principles given the important knowledge gaps that still remain, I would expect at least a discussion of this modelling choice. What would be the aim of the model in the end ? As the plant carbon budget is computed, carbon deficit (or starvation) could for example influences such mortality rates.

**Revised**.

The discussion has been added as the reviewer suggested. "*In the updated model, user-defined mortality rates were used in the simulation. Recent studies showed that the mortality rates at stand level are regulated by various factors, such as the stand age and density, tree basal area, stand squared mean diameter at breast height, standardized precipitation evapotranspiration index, drought length, mean annual temperature and precipitation, elevation and slope (Subedi et al., 2021; Yan et al., 2024). The count-data models combined with random effects of the survey plots have been developed to simulate the tree mortality at different sites (Zhang et al., 2015; Yan et al., 2024). However, the general parameterizations of mortality rates, including the factors of stand, climate and topography, are still not available for the process-oriented models due to knowledge gaps. Thus, in order to predict the carbon cycles of forest using the process-oriented models, more attention should be focused on the simulation of tree mortality under varied environmental conditions and disturbances.*" (**See lines 564-573 in the revised manuscript**).

308: which parameters were calibrated and how? Unless I missed something this important information is missing.

**Revised**.

The statements about model calibration have been added. "*The required parameters for forest simulation, including forest type, carbon contents of leaf and stem and some of eco-physiological parameters, were primarily adapted from the field observations provided by the NESDC or from the peer reviewed literatures (Li, 2018; Li, 2019; Fang, 2022). The other eco-physiological parameters referred to the default values (Table 1). The parameter of fraction of leaf nitrogen in Rubisco (p32) was calibrated using the normalized root mean square error (NRMSE) between observed and simulated carbon and water fluxes during 2003–2007. The upper and lower boundaries of the parameter value (p32) were set as twice and half of the default value. The parameter was identified when the value of NRMSE was the minimum.*" (**See lines 326-332 in the revised manuscript**).

318-320: which parameters were drawn from literature or field observations, and which were calibrated?

**Revised**.

The sources of the eco-physiological parameters have been marked in the revised

Table 1, indicating the values from literatures or calibration, as well as the range of parameters used for calibration. (**See Table 1 in the revised manuscript**).

320: is a soil depth of 1.5 relevant for all three sites? Do you have any information on soil and root depths at these sites?
**Revised**.
The sentence has been revised with references to make it clear. "*The simulated soil profile (0−1.5 m in depth), with the last layer set as rock, was divided into 16 layers and the layer thicknesses were 0.05, 0.1 and 0.5 cm for the 0–0.5, 0.5–1 and 1–1.5 m depths, respectively, according to the previous studies (Guan et al., 2006; Zeng et al., 2008; Zhou et al., 2013).*" (**See lines 333-335 in the revised manuscript**).

324: how long is it to run 13 years of simulations? What were the spatial and temporal resolutions of those simulations. How were the climate data used for this spin-up?
**Revised**.
The details about resolutions and climate data have been added in the revised manuscript. For the site scale simulations of three forests, the spin-up take no more than 5 minutes. "*The simulations of three forest sites were done with the temporal resolution of 3h at the site scale.*" (**See lines 335-336 in the revised manuscript**). "*The climate data used for model spin-up were obtained from the China meteorological forcing dataset (1979–2018) (https://data.tpdc.ac.cn).*" (**See lines 342-343 in the revised manuscript**).

461: it is not crystal clear to me how the different variables can be substantially improved at the daily scale, but not at the annual scale (cf my similar comment on l.35). Can you explain this?
**Revised**.
The explanations have been added as the reviewer suggested. "*Although the simulated ET at the daily scale by the original model showed significant deviations with much more intensive variations, especially for sites of QYZ and DHM, the trade-off effects of extreme values for the original models led to comparable performances of both models in simulating annual ET (Fig. 2j−l).*" (**See lines 458-461 in the revised manuscript**).

496-497: ok but in absence of details on your calibration approach, it is unclear if your model and study is not in a similar situation with eddy flux data: can the model produce reasonable predictions in sites without long-term eddy flux data for calibration ? It would be great to answer this question. How transferable is the model in different sites/environmental conditions? How important are the calibration steps for predictions?
**Revised**.
The statements about model calibration have been added as mentioned above. The discussions about the reasonable prediction of the updated model have been added as reviewer suggested. "*According to Table 1, except for the CBM site with sufficient*

*localized parameters, the eco-physical parameters in the other two sites primarily came from default values. The comparable model performances at the three sites indicated the applicability of the updated model without various observations and comprehensive calibration. But the localization of eco-physical parameters can improve the model performances without doubt.*" (**See lines 561-564 in the revised manuscript**).

510: ok but is such satellite data reliable/good enough to quantify the "subtle changes in leaf phenology" (l. 507) at mote forest sites?

**Revised**.

The sentences have been rewritten to avoid confusion. "*The worst performance for the forest at the DHM site may be attributed to the errors in simulating the photosynthesis of EBT due to the difficulty in modeling the subtle changes in the leaf phenology. Such a difficulty has also been encountered by previous studies, which might be solved by incorporating new mechanisms derived from observations and integrating more complete environmental regulations to vegetation production (Raczka et al., 2013; Yuan et al., 2014). In addition, previous study has showed that assimilating satellite data, e.g., LAI, can significantly improve the performance of process-oriented models in simulating the spatial patters of daily GPP (Yan et al., 2016), which may provide a solution to improve the ability of the modified CNMM-DNDC for simulating the spatial and temporal dynamics of forest GPP at large scales.*" (**See lines 514-521 in the revised manuscript**).

520: it would be interested to include Q10 in your sensitivity analysis then. Why not?

**Revised**.

The sensitivity analysis of $Q_{10}$ in the process of maintenance respiration has been added as the reviewer suggested. "*In updated growth module, the $Q_{10}$ was used for the calculation of maintenance respiration, which not only was a component of ER but also affected the photosynthesis directly. The OAT sensitivity analysis showed that the simulated GPP was more sensitive to $Q_{10}$ than the simulated ER which was also contributed by growth respiration and soil heterotrophic. In addition, the sensitivity index of $Q_{10}$ was higher at the CBM due to the high latitude, which is consistent with the field observation (Yu et al., 2008; Zhang et al., 2019a).*" (**See lines 533-538 in the revised manuscript**).

552: actually, it may be relevant to use different values of SLA for sun and shade leaves given the high plasticity of this trait along light gradient (Niinemets et al. 2015). This could be discussed as well.

**Revised**.

The discussion has been added in view of the reference recommended by the reviewer. "*Previous study also found that light gradients within-canopy substantially affect photosynthetic productivity of leaves influenced by the different combinations of structural, chemical and physiological traits, such as SLA (Niinemets et al., 2015). Maybe using different values of SLA for sun and shade leaves along light gradients*

*could improve the simulation of photosynthesis in future model studies.*" (**See lines 586-589 in the revised manuscript**).

References:

Walker, A. P., Beckerman, A. P., Gu, L., Kattge, J., Cernusak, L. A., Domingues, T. F., ... & Woodward, F. I. (2014). The relationship of leaf photosynthetic traits–Vcmax and Jmax–to leaf nitrogen, leaf phosphorus, and specific leaf area: a meta-analysis and modeling study. Ecology and evolution, 4(16), 3218-3235.

Domingues, T. F., Meir, P., Feldpausch, T. R., Saiz, G., Veenendaal, E. M., Schrodt, F., ... & Lloyd, J. O. N. (2010). Co-limitation of photosynthetic capacity by nitrogen and phosphorus in West Africa woodlands. Plant, Cell & Environment, 33(6), 959-980.

Niinemets, Ü., Keenan, T. F., & Hallik, L. (2015). A worldwide analysis of within-canopy variations in leaf structural, chemical and physiological traits across plant functional types. New Phytologist, 205(3), 973-993.

**Revised**.

The above references have been added in the revised manuscript supporting the results and discussions.

---

## Author Comment (AC2)

Zhang et al. describe an increase in process representation in CNMM-DNDC model. Schemes for allocation, respiration, and mortality were added. The model was set up to run at three forest sites. Model outputs were compared to eddy-covariance data. The model was calibrated and validated, and a sensitivity analysis was performed. My primary concern about this analysis is that I did not perceive a substantial contribution to modelling science. I welcome clarification from the authors, but on my reading of the manuscript, I did not see new concepts, ideas, or methods.

**Revised**.

The improvements of forest growth module for the CNMM-DNDC would be a valuable contribution to the corresponding modeling community which has been stated in the revised manuscript. "*To achieve such improvements is a valuable contribution to the corresponding modelling community. Firstly, as the CNMM-DNDC aimed at modelling or predicting of multiple ecosystem variables (e.g., emissions/uptakes of carbon and/or nitrogen gases from terrestrial ecosystems, evapotranspiration, productivity, soil erosion, and slope runoff and catchment discharges of particle and/or dissolved carbon, nitrogen and phosphorous at plot, ecosystem, landscape, catchment/river basin, regional or global scales) concerned in the implementation of the United Nations Sustainable Development Goals (SDGs) by 2030, the shortcomings in simulating the biogeochemical processes of forest ecosystems would hinder the effective application of the model. Secondly, the reliable CNMM-DNDC model with new growth module would have the potential to study the interactions between forest carbon pools and hydrological processes, such as the losing of soil organic carbon due to the thawing of permafrost, which has attracted more attentions under climate change.*" (**See lines 107-116 in the revised manuscript**).

Beyond the issue of the significance, I think that several other points would need to be addressed:

1) The general description of CNMM-DNDC in section 2.1.1 can be improved. I would like to know what the state variables are, what kinds of equations govern the state variables, and what kinds of boundary conditions, initial conditions, and forcing the model requires.

**Revised**.

The equations, state variables and parameters related to the carbon and nitrogen cycling has been added in Tables S1-S3 of the supplementary materials (**See Table S1-S3 in the revised supplementary materials**). The requirements for the simulation, including boundary conditions, initial conditions and forcing, have been detailed as reviewer suggested. "*The data required for the simulation include land use type, soil properties of individual layers (soil organic carbon, total nitrogen, clay content, bulk density, pH, etc.), meteorological forcing (hourly air temperature, precipitation, wind speed, solar radiation, etc.), biological data (plant type, nitrogen content, plant height, root depth, etc.), initial conditions (soil depth, soil temperature, soil moisture, annual amounts of dry and wet nitrogen deposition etc.), management practices (start and*

*end dates, methods and/or amounts of individual management practices including tillage, fertilization, irrigation and flooding for croplands), and boundary data (start date, period, time step of simulation, depth of soil profile etc.).*" (**See lines 136-142 in the revised manuscript**).

2) I think that Equation 21 has an error. I propose that the numerator should be 1 - f_HR - R_{soilCN} / R_{litCN}. Is this a typo? A bug in the code?
**Revised**.
The equation has been revised. It is a typo, but not a bug in the code (**See Eq.21 in the revised manuscript**).

3) I would be interested in a description of N cycle inputs (fixation, deposition, etc.) and outputs (gas losses, leaching, etc.). Are values for these quantities known at the study sites?
**Revised**.
The descriptions about nitrogen fixation and deposition have been added as reviewer suggested. "*The nitrogen fixation was considered using the default value of 0.0004 kg $m^{-2}$ $y^{-1}$ in Biome-BGC during the simulation (Fang, 2022).*" (**See lines 332-333 in the revised manuscript**). "*......as well as the nitrogen deposition, were primarily obtained from the National Ecosystem Science Data Center (NESDC; https://www.nesdc.org.cn/). Based on the annual amounts of dry and wet nitrogen deposition (Jia et al., 2019; 2021), the daily dry nitrogen deposition and the nitrogen concentration in wet deposition were calculated as model driving.*" (**See lines 317-320 in the revised manuscript**).

4) More information needs to be provided on how the model calibration was carried out. What kinds of algorithms were used? How many simulations were done? How was convergence assessed?
**Revised.**
The statements about model calibration have been added. "*The required parameters for forest simulation, including forest type, carbon contents of leaf and stem and some of eco-physiological parameters, were primarily adapted from the field observations provided by the NESDC or from the peer reviewed literatures (Li, 2018; Li, 2019; Fang, 2022). The other eco-physiological parameters referred to the default values (Table 1). The parameter of fraction of leaf nitrogen in Rubisco (p32) was calibrated using the normalized root mean square error (NRMSE) between observed and simulated carbon and water fluxes during 2003–2007. The upper and lower boundaries of the parameter value (p32) were set as twice and half of the default value. The parameter was identified when the value of NRMSE was the minimum.*" (**See lines 326-332 in the revised manuscript**).The sources of the eco-physiological parameters have been marked in the revised Table 1, indicating the values from literatures or calibration, as well as the range of parameters used for calibration. (**See Table 1 in the revised manuscript**).

5) Line 323: The N cycle can take much longer than 13 years to equilibrate (Thornton and Rosenbloom 2005, Ecological Modeling, 189, 25-48). In what sense is the 13 year spin-up really satisfactory? Is the N cycle still far from equilibrium?

**Reponses**.

The details have been added to make it clear. "*In this study, the soil carbon and nitrogen pools were initialized by the observed data. The initial state of forest was constrained by the carbon contents of leaf and stem based on the observed biomass, as well as the proportion among different organs, which was not the model's native dynamics (Table S5; Thornton and Rosenbloom, 2005).*" (**See lines 336-339 in the revised manuscript**)

6) The OAT sensitivity analysis is problematic. From the Discussion section, the authors seem to be aware that it is problematic. As things stand in the manuscript, I do not have confidence in the sensitivity analysis results. It would be improvement to show scatterplots so that the linearity of the response could be assessed, but that still wouldn't solve the problem of parameter interactions. Using something like the Morris method would address these issues, and wouldn't really require more iterations.

**Revised**.

Both the OAT and Morris method has been applied for the sensitivity analysis. Monte Carlo Monte Carlo test with Latin hypercube sampling was used in the Morris method with the iterations of 2000. (**See lines 371-387 in the revised manuscript**). "*The global sensitivity analysis using Morris method (Fig 5) showed similar results at the CBM site with selected sensitive parameters of SLA (p30), FLNR (p32) and carbon allocation rate of new fine root to new leaf (p10). Annual GPP and ER fluxes at the QYZ site were sensitive to SLA (p30), FLNR (p32) and LFRT (p6) using the both methods of OAT and Morris. However, only canopy light extinction coefficient (p28) was identified as a sensitive parameter at the DHM site, which was not identical to the results of OAT.*" (**See lines 483-487 in the revised manuscript**). "*For the results of global analysis, SLA, FLNR and LFRT were also identified as sensitive parameters. But the effects of carbon allocation rate of new fine root to new leaf or canopy light extinction coefficient on annual GPP and ER could not be ignored at the site of CBM or DHM, respectively. Canopy light extinction coefficient determines the amount of absorbed photosynthetically active radiation and thus regulates the GPP and ER (White et al., 2000). The analysis of eco-physiological parameters suggests that the sensitive parameters may be consistently influential, independent of sites or the type of sensitivity analysis. But the ranking of the parameters may vary according to specific species and regions (Raj et al., 2014).*" (**See lines 605-611 in the revised manuscript**). "*Thus, global sensitivity analysis using the Morris method was also applied which can reflect the interactive effects of changes in multiple parameters/inputs (Odongo et al., 2013; Raj et al., 2014). The results of both methods were not totally consistent for three sites, which proved the limitations of OAT and necessity of global sensitivity analysis considering the comprehensive effects of*

*multiple parameters/inputs (Saltelli et al., 2000; Odongo et al., 2013)."* (**See lines 636-639 in the revised manuscript**).

7) I would have liked to have seen a discussion of whether the fitted parameter values were reasonable, and whether the variation across sites made sense in terms of basic biology.

**Revised**.

The calibration and sources of eco-physiological parameters have been detailed as mentioned above. The discussion about parameters among three sites has been added as reviewer suggested. *"The eco-physical parameters used in this study are comparable with those measured or calibrated in other studies (Tables S8−S9; Li, 2018; Li, 2019; Fang, 2022). Due to the limited studies at the QYZ site, the required eco-physical parameters, excluding the calibrated one, were directly derived from those at the DHM site. The values of FLNR showed increased tendency from low latitude to high latitude, supporting the vigorous growth of trees during growing season at the CBM site (Li, 2019). Li (2018) found that the parameters of carbon allocation rate of new fine root to new leaf (p10), carbon allocation rate of new stem to new leaf (p11) and vapour pressure deficit for the start of conductance reduction (p36) show high spatial heterogeneity for DBT and EBT. In this study, the above three parameters also varied along the latitude."* (**See lines 554-560 in the revised manuscript**).

8) To me, it is a problem that error bars are almost entirely missing. What is the uncertainty in model predictions?

**Revised**.

The model simulation error has been calculated using the model relative biases during model validation and Monte Carlo Monte Carlo test with Latin hypercube sampling, which has been presented in the Text S3 and Figure 2 (**See Text S3 and Figure 2 in the revised supplementary materials and manuscript**).

9) Model development focused on things like allocation and mortality, yet there was no validation of observables related to allocation and mortality. If allocation and mortality are things that are added to the model, the authors should present direct evidence that these schemes are producing acceptable results (for example: comparison of observed and simulated wood growth; comparison of observed and simulated mortality; etc.).

**Reponses**

For the original model, allocation was considered only for aboveground biomass and belowground biomass without the transportation among different organs or tissues, which is not suitable for trees. Therefore, the new growth module was introduced to perfect the scientific processes of the model. The observed fluxes of carbon and water were used for model validation, but the observed data of allocation and mortality were not available. *"Due to the limited available observations at the three sites, validations*

*of allocation and mortality, which were newly introduced key processes, were not available in this study.*" (**See lines 310-312 in the revised manuscript**).

10) English usage throughout the manuscript is problematic. The manuscript needs to be proofread for grammar and style.
**Revised**.
The English usage has been revised throughout the manuscript.

Technical comments:
1. What is a "humad" (line 174)?
**Reponses**.
In the model, the "humads" indicates the liable humus, while the "humus" indicates the resistant humus, which is derived from the biogeochemical model of DNDC. The decomposition rates of above two components are different. "*The humads and humus defined in the DNDC indicate the liable humus and resistant humus with different decomposition rates, respectively.*" (**See lines 182-183 in the revised manuscript**).

2. How does pH affect the model?
**Reponses**.
The statement has been added to make it clear. "*In the model, pH short-term variations after urea application for uplands and paddy fields and soil acidification after tea plantation can be simulated.*" (**See lines 154-155 in the revised manuscript**).

3. I think it would be nice to have Table S3 in the main text.
**Revised**.
The table has been added in the main text as the reviewer suggested. (**See Table 1 in the revised manuscript**).